# Label-free adaptive optics single-molecule localization microscopy for whole zebrafish

**Sanghyeon Park** [1,2,7], **Yonghyeon Jo** [1,2,7], **Minsu Kang**[3], **Jin Hee Hong** [1], **Sangyoon Ko**[3], **Suhyun Kim** [4], **Sangjun Park**[5,6], **Hae Chul Park** [4], **Sang-Hee Shim** [3] ✉ **& Wonshik Choi** [1,2] ✉

Specimen-induced aberration has been a major factor limiting the imaging depth of single-molecule localization microscopy (SMLM). Here, we report the application of label-free wavefront sensing adaptive optics to SMLM for deep-tissue super-resolution imaging. The proposed system measures complex tissue aberrations from intrinsic reflectance rather than fluorescence emission and physically corrects the wavefront distortion more than three-fold stronger than the previous limit. This enables us to resolve sub-diffraction morphologies of cilia and oligodendrocytes in whole zebrafish as well as dendritic spines in thick mouse brain tissues at the depth of up to 102 μm with localization number enhancement by up to 37 times and localization precision comparable to aberration-free samples. The proposed approach can expand the application range of SMLM to whole zebrafish that cause the loss of localization number owing to severe tissue aberrations.

Single-molecule localization microscopy (SMLM) improves the spatial resolution of a diffraction-limited fluorescence microscope by more than an order of magnitude[1,2]. The approach has widely been used in diverse biological studies owing to its simplicity and high resolution[3,4]. However, its working depth is much shallower than those of diffraction-limited microscopy. This is because single-molecule localization is highly susceptible to sample-induced scattering and aberration. In fact, tissue scattering is either less problematic in relatively transparent organisms or can be effectively reduced by tissue clearing[5]. However, even relatively weak sample-induced aberration for diffraction-limited imaging modalities can have detrimental effects on SMLM imaging. More specifically, the point spread functions (PSF) blur caused by tissue aberration reduces the number of photons detected at each camera pixel. This results in signal-to-noise ratio (SNR) reduction and loss of localizations[6,7]. In addition, tissue aberration even distorts PSF shape, which causes erroneous localization. It leads to additional loss of localization and degradation in localization precision.

Adaptive optics (AO) provides a suitable solution for these problems. Literally, AO actively controls PSF with wavefront shaping devices such as deformable mirrors and spatial light modulators (SLM)[8]. AO was first introduced in astronomy to deal with PSF distortion resulting from atmospheric turbulence[9]. Because similar issues exist in bioimaging in which complex tissue structures distort the wavefront, AO also has been applied to microscopy. Especially, AO has recently been implemented in diverse super-resolution fluorescence microscopy methods such as stimulated emission depletion (STED) microscopy[10,11], structured illumination microscopy (SIM)[12], and SMLM[13,14]. AO super-resolution imaging approaches can largely be categorized into two types: wavefront-sensing AO and sensorless AO. In wavefront-sensing AO, the wavefront of the emission beam is directly measured from either artificial[15] or intrinsic[16] guide stars with a Shack-Hartmann wavefront sensor. Two-photon fluorescence emissions have often been used as guide stars without using fluorescent particles. However, this approach has not yet been implemented in

[1]Center for Molecular Spectroscopy and Dynamics, Institute for Basic Science, Seoul, Republic of Korea. [2]Department of Physics, Korea University, Seoul, Republic of Korea. [3]Department of Chemistry, Korea University, Seoul, Republic of Korea. [4]Department of Biomedical Sciences, Korea University, Ansan, Republic of Korea. [5]Department of Medical Life Sciences, College of Medicine, The Catholic University of Korea, Seoul, Republic of Korea. [6]Department of Biomedicine and Health Sciences, The Catholic University of Korea, Seoul, Republic of Korea. [7]These authors contributed equally: Sanghyeon Park, Yonghyeon Jo. ✉e-mail: sangheeshim@korea.ac.kr; wonshik@korea.ac.kr

SMLM probably because single-molecule signals are too weak as guide stars for wavefront measurement.

On the contrary, sensorless AO has been widely used in SMLM. In this approach, wavefront shaping devices are controlled for optimizing elaborately devised image quality metrics of SMLM images[13,14,17,18]. So far, deformable mirrors have been used to determine a specific amplitude of each Zernike mode that maximizes image metrics. These approaches enable successful SMLM imaging within cells and relatively thin tissue slices[13,14,17]. However, the requirement for recording single-molecule blinking images in each optimization step imposes a few constraints. First, aberrations should be mild enough to detect single-molecule PSFs. Otherwise, it is impossible to evaluate image quality metrics, which is essential for initiating optimization processes. Second, each iteration step consumes single-molecule images for optimization and takes some time, during which photobleaching occurs. Third, the optimization process is typically nonlinear; its efficiency is highly dependent on the choice of image quality metrics and optimization methods[18]. All these constraints preclude the correction of high-order aberration, thereby limiting achievable imaging depth in SMLM. In fact, most of the previous AO-SMLM modalities handled mild aberrations whose root-mean-square (RMS) wavefront distortion is less than 1 rad even for in vitro assays with artificial aberration[18]. Therefore, the main benefit of previous AO modalities is improving image contrast rather than fully reconstructing unseen structures[13,19]. Consequently, imaging depth in AO-SMLM has still been only a couple of tens of microns.

Closed-loop accumulation of single-scattering (CLASS) microscopy[20,21] can be a suitable solution for overcoming the major limitations of previously presented AO-SMLM methods. CLASS microscopy records multiple interferometric reflectance images from tissue structures at many different illumination angles. Its algorithm finds a sample-induced aberration based on the reflection matrix constructed from measured reflectance images. Since CLASS does not rely on single-molecule PSF images, no bleaching occurs during aberration measurements. Furthermore, it can find aberrations even when single-molecule PSFs are completely invisible due to complex aberrations.

In this study, we employ CLASS to SMLM for super-resolution imaging deep within the whole zebrafish. Using CLASS, we identified tissue aberration from the label-free measurement of the intrinsic reflectance signal of the tissue where single-molecule fluorescence was too weak for the detection or too aberrant for precise localization. By physically correcting aberrations with an SLM in the emission beam path of SMLM, abnormal PSFs were restored to near-ideal PSFs. This led to the reduction in PSF width and enhancement of the SNR in SMLM imaging in which the centroids of individual single-molecule PSFs were fitted while they were randomly photoswitched. In doing so, we realized SMLM imaging of whole zebrafish larvae at a depth of up to 102 μm. In whole zebrafish, the aberration had RMS wavefront distortion of 2.13–3.08 rad, which is ~2–3 times stronger than the 1 rad limit of previous AO-SMLM methods[18]. Our proposed AO-SMLM was able to correct this level of aberration and made a significant improvement in the localization number, up to 37.4-fold enhancement (corresponding to ~6.12-fold Nyquist resolution enhancement), compared to the typical 2–8 times enhancement seen in previous studies[13,18]. Here, localization number means the number of localized PSFs identified by the PSF fitting algorithm. In doing so, we could improve localization precision up to 3.61-fold. Essentially, our system enables super-resolution imaging of whole zebrafish with localization precisions close to those of aberration-free cells. We demonstrated resolving various sub-diffraction structures formed within thick brain tissues and whole zebrafish larvae which were either completely invisible or obscured without AO. Specifically, we resolved dendritic spines in mouse brain tissues as well as ciliary membranes and oligodendrocyte membranes in the hindbrain and spinal cords of whole zebrafish larvae.

## Results

### Label-free AO-SMLM setup

Our AO-SMLM system was built on a commercial inverted microscope (Fig. 1a and Supplementary Fig. 7 for detailed layout). One port of the microscope was connected to the CLASS microscope (yellow box in Fig. 1a), and the other to the SMLM (cyan box in Fig. 1a). Both microscopes shared a sample, an objective lens, and a tube lens (not shown in Fig. 1a). A built-in flip mirror allows switching between the two microscope systems. The CLASS microscope was equipped with a superluminal laser diode whose center wavelength (678 nm) was near the emission peak wavelength of Alexa Fluor 647 used for SMLM. The CLASS microscope recorded multiple interference images of intrinsic reflection from the sample at many different illumination angles. Based

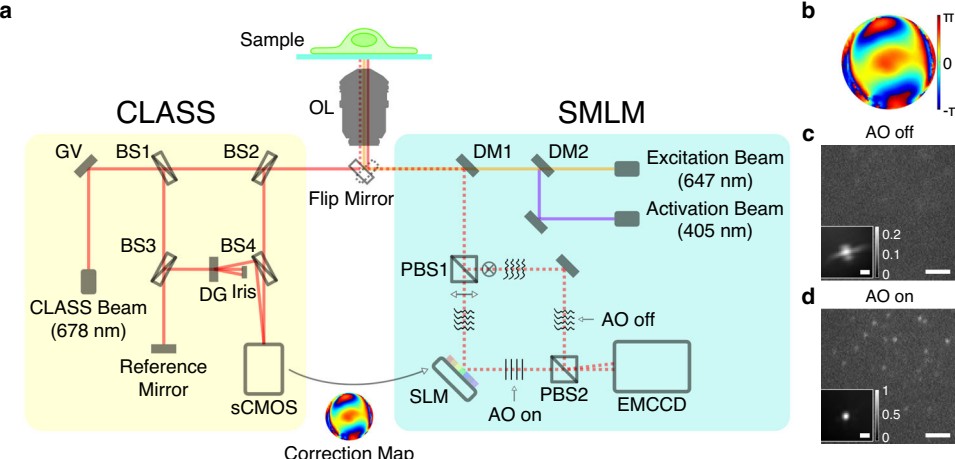

**Fig. 1 | Experimental setup. a** Simplified layout of the experimental setup composed of the CLASS microscope (yellow box) and SMLM (cyan box). OL objective lens, GV two-axis galvanometer mirror, DG diffraction grating, BS1-4 beam splitters, DM1-2 dichroic mirrors, PBS1-2 polarizing beam splitters, SLM spatial light modulator, and gray rectangles without labels: mirrors. **b** Aberration correction map whose radius is 1.2/λ in spatial frequency where λ is the emission peak wavelength of Alexa Fluor 647. **c, d** Single-frame raw images of single-molecule PSFs simultaneously recorded without (**c**) and with (**d**) AO, respectively. Images are normalized with respect to AO on. Insets show ensemble-averaged normalized PSFs of the first 10,000 frames. Scale bars indicate 2.5 μm and 500 nm (insets). Source data are provided as a Source Data file.

on these images, the sample-induced aberration was identified by applying an algorithm that maximizes the single-scattering intensity (see "Detailed process of CLASS algorithm" section in Supplementary Note 1 for the detailed process).

The SMLM setup was equipped with an excitation source (647 nm) and an activation source (405 nm). The fluorescence emission (dotted lines in Fig. 1a) captured by the objective lens (1.2 NA, water immersion lens) was split into two branches by a polarizing beam splitter (PBS1 in Fig. 1a). In the horizontally polarized beam path, we installed an SLM at a plane conjugate to the pupil plane of the objective lens. To correct aberration, we displayed the opposite phase of the CLASS-evaluated aberration map (Fig. 1b) with the correction map of pre-measured SMLM setup aberration (Supplementary Fig. 11). Here, the size, position, and orientation of the aberration correction map displayed on the SLM were calibrated prior to CLASS (Supplementary Figs. 9 and 10). Being reflected from the SLM, the horizontally polarized emission beam was corrected and finally arrived at one corner of the camera sensor (EMCCD). The other emission beam with vertical polarization was directly sent to another corner of the camera sensor without passing through the SLM. This enabled simultaneous acquisition of aberration-uncorrected (AO off in Fig. 1a) and -corrected (AO on in Fig. 1a) single-molecule blinking images with a single camera. Photon loss due to the reflection from the SLM was negligible (Supplementary Fig. 8). This justifies the final step of CLASS-SMLM imaging process (see Supplementary Fig. 12), i.e., fair comparison between AO-off and -on SMLM images.

The effect of AO is revealed by comparing simultaneously acquired snapshots of single-molecule images (Fig. 1c, d and Supplementary Movie 1). Especially, ensemble-averaged PSFs show successful recovery of highly distorted PSF (insets in Fig. 1c, d). With AO, the FWHM (full width at half maximum) of the ensemble-averaged PSF decreased from 1430 to 380 nm (Supplementary Table 1), which is very close to the PSF width of the residual system aberration (Supplementary Fig. 11). This indicates that all specimen-induced aberration was successfully removed. In addition, the Strehl ratio (i.e., peak intensity ratio of aberrated PSF to ideal PSF) was improved by 4.27 times.

It is worth noting that the abnormal PSFs were restored to near-ideal PSFs for all the molecules within a certain area known as the isoplanatic patch. The isoplanatic patch is an area within which aberration stays the same. Its size depends on the type and internal structure of the sample. In the case of brain tissues, it was larger than the FOV of the SMLM imaging (~33 × 33 μm²). In the case of zebrafish, it was as small as ~10 × 10 μm². Therefore, we segmented the FOV and evaluated aberrations in each of ~10 × 10 μm² subarea. For the acquisition of the CLASS-SMLM image, we chose one of the subareas where the structures of interest are located and applied the aberration correction for the corresponding subarea. PSF correction is independent of time as the sample was fixed during the entire imaging session.

### Proof-of-concept imaging of microtubules in a cell through an aberrating layer

To assess the performance of our AO-SMLM, we imaged microtubules immunolabeled with Alexa Fluor 647 in a COS-7 cell. Cells were cultured on cover glasses onto which 100-nm-diameter gold nanoparticles were attached before plating cells. For inducing severe aberration, an artificial aberration layer was inserted between the objective lens and the cover glass. Its aberration was measured via interferometric reflectance imaging of the gold nanoparticles with CLASS microscopy (inset in Fig. 1c: tilt and defocus Zernike modes removed). As demonstrated for imaging whole zebrafish and thick brain tissues, gold nanoparticles were unnecessary for deep-tissue imaging because CLASS microscopy exploits intrinsic reflection signals from inhomogeneous tissue structures to identify aberration (Supplementary Fig. 5). In fact, CLASS microscopy is so sensitive that

even weak intrinsic reflection signals from brain tissues under the mouse skull are strong enough to measure the aberrations[22,23].

The identified aberration has the RMS wavefront distortion of 2.22 rad, which is more than twice greater than the 1 rad limit of precedent AO-SMLM approaches. Aberrations at this level require correction of at least the first 100 Zernike modes (Supplementary Fig. 1). This is far beyond the capacity of previous AO-SMLM studies in which only the first ~20 Zernike modes were controlled at most[13,14,18,19]. Successful correction of this high-order aberration is attributed to the precise aberration measurement via CLASS and an SLM's higher correction resolution than that of a deformable mirror.

Next, we compared AO-off and -on images to check the effect of this severe aberration. In diffraction-limited fluorescence images, we observed evident improvement in image intensity and sharpness (Fig. 2a, b). Without AO, images were quite blurred while individual microtubules were discernible with AO. Surprisingly, SMLM images showed much more dramatic differences between AO off and on (Fig. 2c, d). Without AO, only rough shapes of microtubules were barely seen. With AO, however, individual microtubules were clearly resolved. The magnified views of two different regions (yellow and green boxes in Fig. 2c, d) showed this difference more evidently (Fig. 2e–h). Without AO, tubular structures of microtubules were completely invisible due to insufficient localization number (Fig. 2e, g). Tubular structures appeared only with AO as demonstrated in the cross-sectional profiles for the white boxes in Fig. 2f, h (Fig. 2i, k). In a region with a single isolated microtubule (yellow box) as well as another region with densely populated microtubules (green box), the width of each microtubule was not measurable at all without AO. With AO, however, FWHMs of the microtubules were successfully measured in both regions. In the yellow boxes, the FWHM of a microtubule was measured as 69 nm (Fig. 2i). Considering the microtubule diameter (~25 nm) and size of primary and secondary antibodies (10–15 nm), this value agrees with well-known widths of microtubules[24,25]. In the green boxes, two microtubules separated by 170 nm were clearly resolved. Their FWHMs were measured as 82 and 85 nm, respectively (Fig. 2k). A more detailed analysis quantifies the enhancements in localization number and localization precision (Supplementary Fig. 2).

According to the nearest neighbor analysis[26] (Fig. 2j, l; see "SMLM analysis" section in "Methods"), localization precision was improved by 2.32–3.61 times to the value on par with that from an aberration-free cell (Supplementary Fig. 3). Furthermore, AO increased localization number by 14.2–25.5 times. This is remarkable compared with previous AO-SMLM studies in which localization numbers increased by ~2–8 times[13,18]. We also conducted a Fourier ring correlation (FRC) analysis[27] estimating the combined effect of localization precision and localization number density, and quantified the resolution improvement from 134 to 41 nm (Supplementary Fig. 4).

### Deep-tissue SMLM imaging in thick mouse brain tissues

Conventional SMLM imaging has suffered from shallow imaging depth of sub-100-μm thickness[14,18,19]. We applied our AO-SMLM to thick (thickness of 150–200 μm) brain slices of Thy1-EGFP transgenic mice. We targeted the dendritic spines of neurons expressing GFP. For SMLM imaging, brain slices were labeled with anti-GFP antibodies conjugated with Alexa Fluor 647 (see "Preparation of mouse brain slices" section in "Methods"). We collected reflectance images from biological structures such as cell bodies, blood vessels, and myelin (Supplementary Fig. 5) for evaluating tissue aberration via CLASS microscopy. Note here that no guide stars, such as gold nanoparticles, were used for CLASS microscopy.

At a depth of 50 μm in a 150-μm-thick mouse brain slice, the identified sample aberration (bottom-left inset in Fig. 3a) had RMS wavefront distortion or 1.37 rad, which is beyond the 1 rad limit of previous AO-SMLM methods. Although this aberration is weaker than that of the cell with an aberration layer in Fig. 2, the obscured

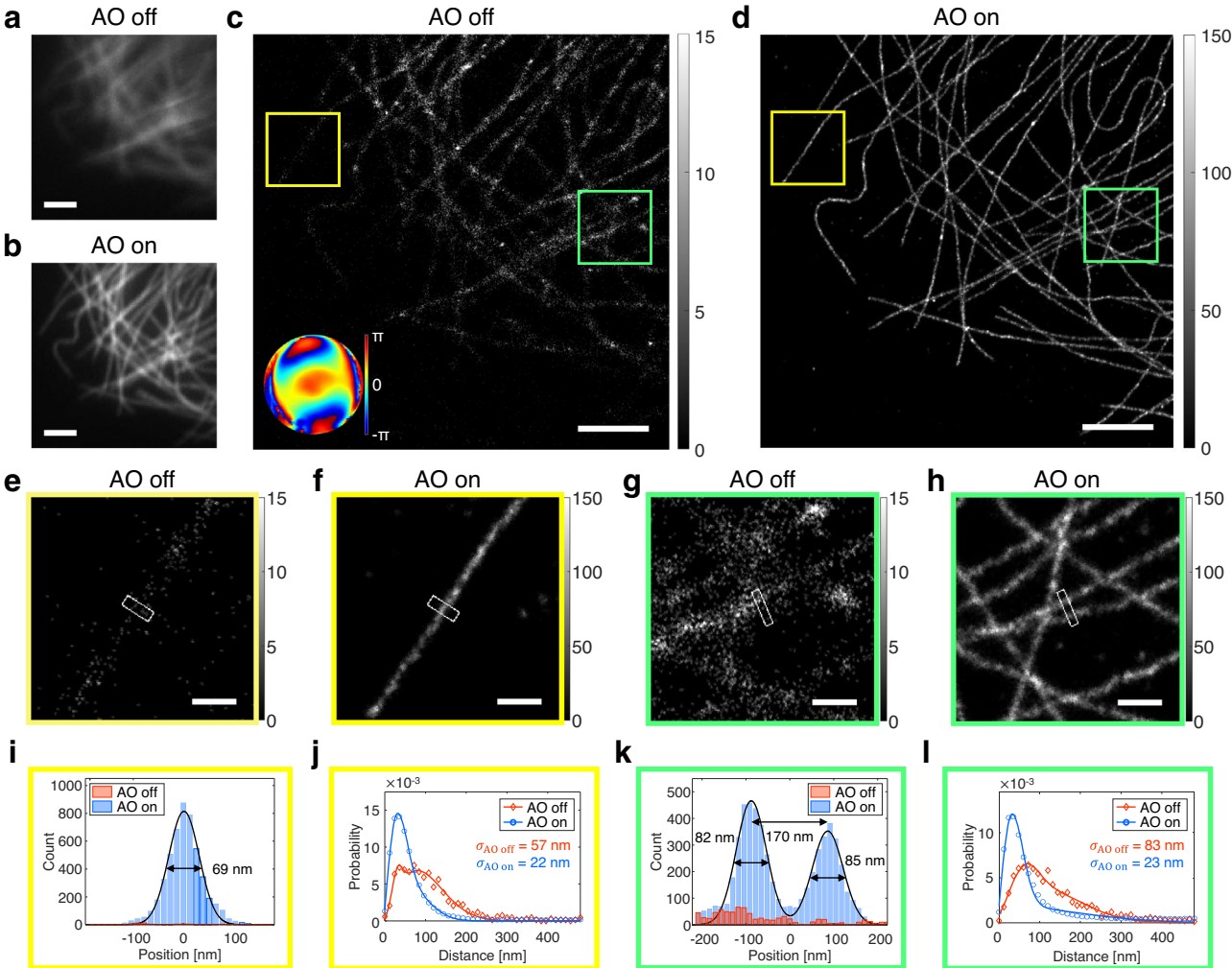

**Fig. 2 | Demonstration of AO-SMLM in a COS-7 cell with an aberration layer.** **a, b** Diffraction-limited fluorescence images without (**a**) and with (**b**) AO, respectively. Images are normalized with respect to AO on. Scale bars indicate 2.5 μm. **c, d** SMLM images without (**c**) and with (**d**) AO, respectively. Inset in (**c**) indicates the aberration correction map. Color bars indicate localization numbers. Scale bars indicate 2.5 μm. **e–h** Magnified views of the yellow and green boxes in (**c, d**), respectively. Color bars indicate localization numbers. Scale bars indicate 500 nm. **i** Cross-sectional profiles of white boxes in (**e, f**). **j** Nearest neighbor analysis results for the yellow boxes in (**c, d**) and other areas where single microtubules were isolated. **k** Cross-sectional profiles of white boxes in (**g, h**). **l** Nearest neighbor analysis results for the green boxes in (**c, d**) and other areas where microtubules were confluent. Source data are provided as a Source Data file.

single-molecule PSF resulted in notable differences in SMLM images. With AO, only thick stems of neural structures were barely visible. In contrast, heads and thin necks of dendritic spines were properly reconstructed only with AO (Fig. 3a). This difference was more clearly revealed in the magnified views of the regions indicated by arrows (Fig. 3b, c). The FWHMs of dendritic spine necks were measured as 89 and 100 nm, respectively. The improvement of SMLM images can be attributed to a 9.32-fold increase of localization number (Fig. 3d) and improvement of localization precision from $\sigma_{\text{AO off}} = 66$ nm to $\sigma_{\text{AO on}} = 38$ nm (Fig. 3e) by AO.

Next, we performed a similar SMLM imaging for a more challenging, thicker sample. At a depth of 74 μm in a 200-μm-thick mouse brain tissue, the measured aberration had RMS wavefront distortion of 0.983 (top-left inset in Fig. 3f). Similar to the previous case in Fig. 3a–c, only thick stems were observed without AO while dendritic spines were well resolved with AO (Fig. 3f). The effect of AO was more evident in the magnified views of the regions indicated by arrows (Fig. 3g–i). The FWHMs of dendritic spine necks were measured as 96, 140, and 130 nm, respectively. Again, this remarkable improvement of SMLM images is due to a 4.79-fold increase of localization number (Fig. 3j) and improvement of localization precision from $\sigma_{\text{AO off}} = 60$ nm to $\sigma_{\text{AO on}} = 37$ nm (Fig. 3k) by AO.

## Deep-tissue SMLM imaging in a whole zebrafish

For decades, zebrafish embryos and larvae have been widely used as model organisms for studying vertebrate gene function and human genetic diseases[28]. However, whole zebrafish larvae have hardly been investigated with super-resolution microscopy due to intense aberrations. Here, we present applications of our AO-SMLM for imaging nanoscale morphology of cilia[29] and oligodendrocytes[30] deep within the hindbrain and near spinal cords of whole zebrafish.

A zebrafish was mounted with its back against the coverslip (Fig. 4a). Various regions of zebrafish larvae were explored with our AO-SMLM (Fig. 4b). At first, we imaged cilia in a whole 3-dpf (days post fertilization) *Tg(bactin2::Arl13b-GFP)* zebrafish with GFP expressed on ciliary membranes. This zebrafish line was selected because it is an important animal model for embryonic development and human diseases such as ciliopathy[31]. For SMLM imaging, the zebrafish was fixed and immunolabelled with anti-GFP antibodies conjugated with Alexa Fluor 647 (see "Preparation of zebrafish" section in "Methods"). At a depth of 82 μm, we obtained SMLM images of several cilia (Fig. 4c–e). Once again, the aberration was identified by CLASS microscopy from intrinsic reflectance images without any artificial guide stars such as gold nanoparticles (Supplementary Fig. 5). The RMS wavefront distortion of the aberration

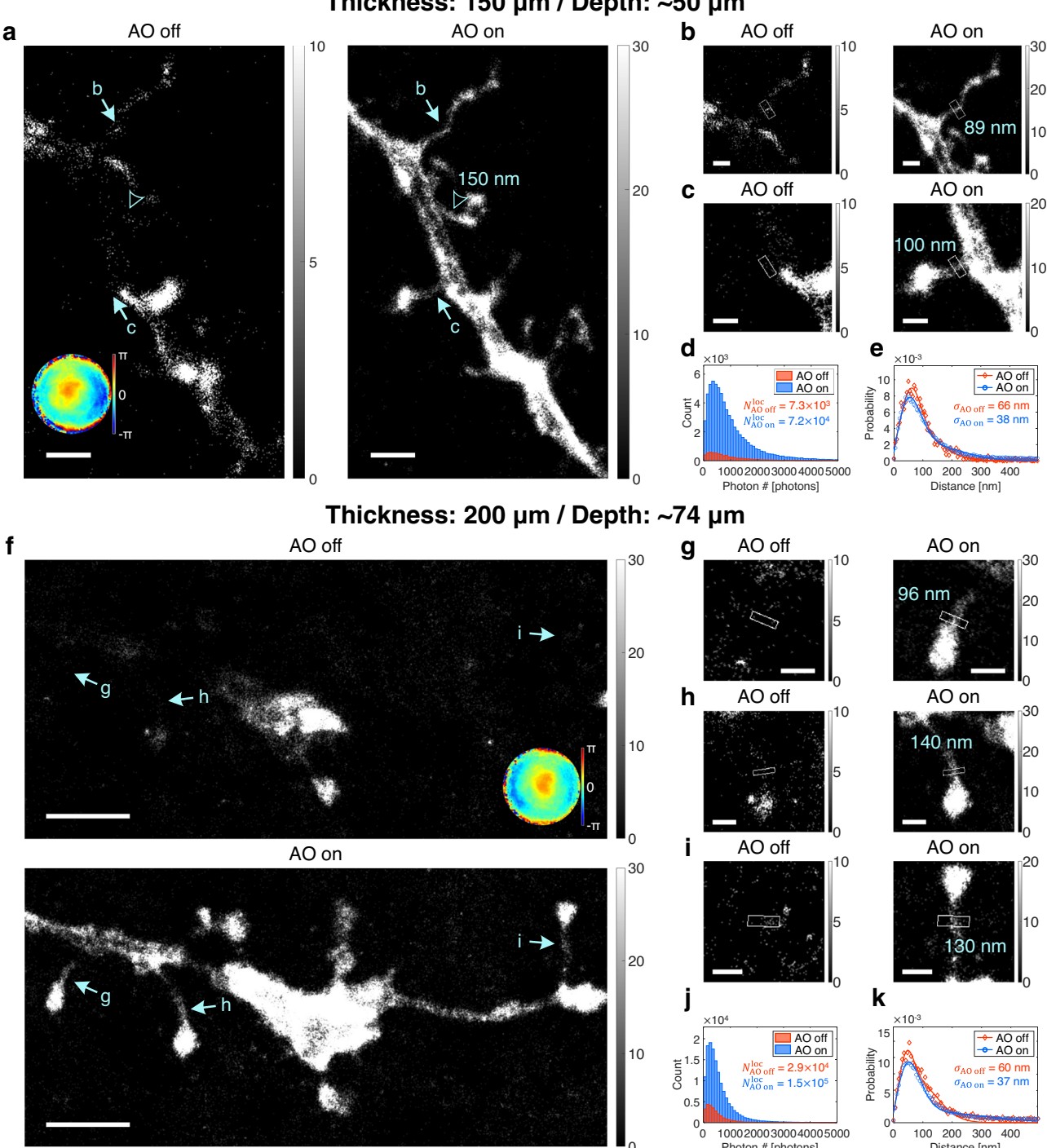

**Fig. 3 | Deep-tissue SMLM images of dendritic spines in mouse brain slices without and with AO. a** SMLM images of dendritic spines at the depth of 50 μm in a 150-μm-thick mouse brain slice without and with AO. Bottom-left inset in AO-off image shows aberration correction map. The FWHM value of a dendritic spine neck is indicated by an arrowhead. Color bars indicate localization numbers. Scale bars indicate 2 μm. **b, c** Magnified views of regions indicated by arrows in (**a**). FWHM values of dendritic spine necks (white boxes) are written in AO-on images. Color bars indicate localization numbers. Scale bars indicate 500 nm. **d, e** Histograms of photon number per emission PSF (**d**) and nearest neighbor analysis results (**e**) of

(**a**). **f** SMLM images of dendritic spines at the depth of 74 μm in a 200-μm-thick mouse brain slice without and with AO. Top-left inset in AO-off image shows aberration correction map. Color bars indicate localization numbers. Scale bars indicate 2.5 μm. **g–i** Magnified views of regions indicated by arrows in (**f**). FWHM values of dendritic spine necks (white boxes) are written in AO-on images. Color bars indicate localization numbers. Scale bars indicate 500 nm. **j, k** Histograms of photon number per emission PSF (**j**) and nearest neighbor analysis results (**k**) of (**f**). Source data are provided as a Source Data file.

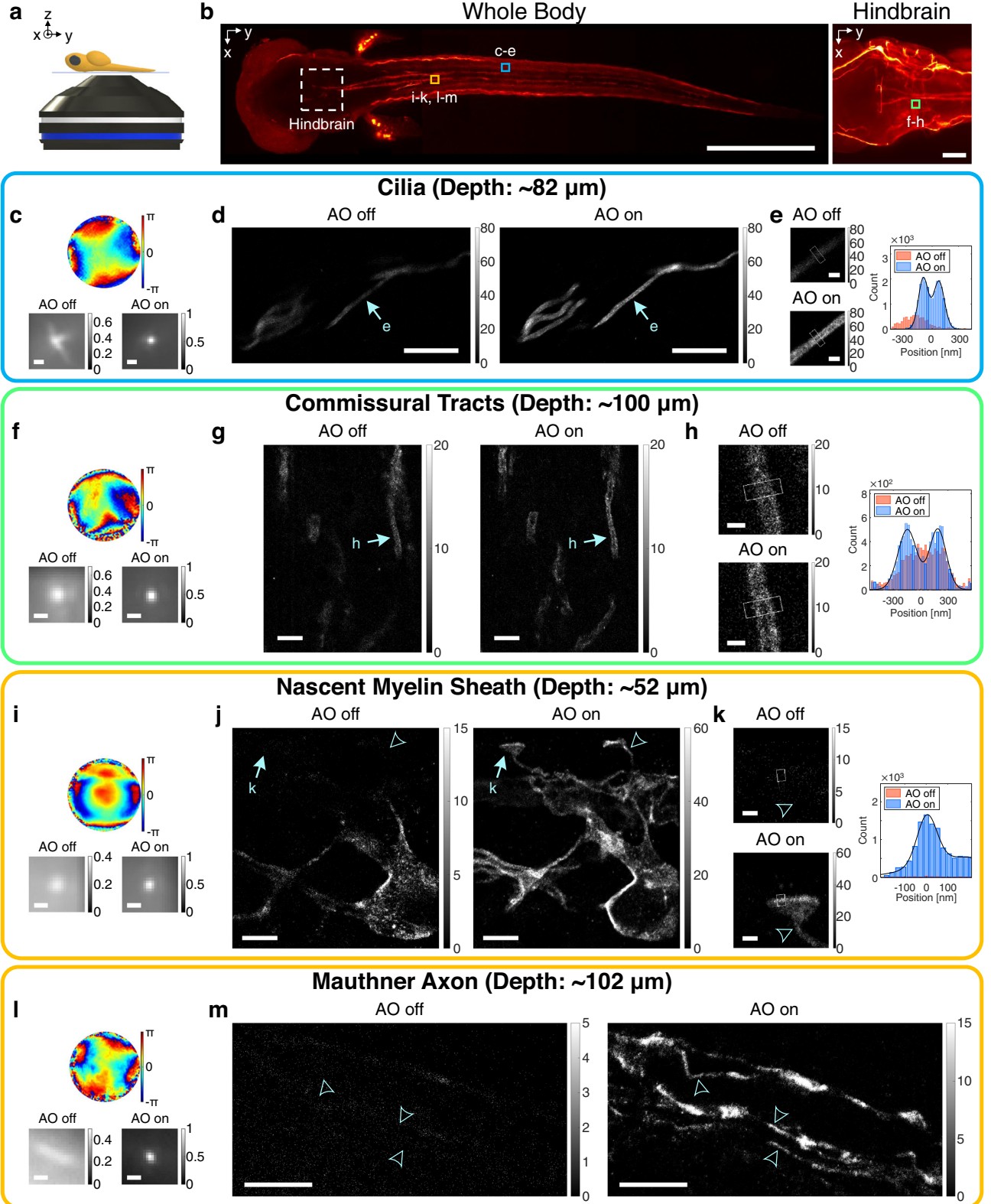

**Cilia (Depth: ~82 μm)**

**Commissural Tracts (Depth: ~100 μm)**

**Nascent Myelin Sheath (Depth: ~52 μm)**

**Mauthner Axon (Depth: ~102 μm)**

(top row in Fig. 2c) was 2.14 rad, which results in distorted single-molecule PSFs (bottom left in Fig. 2c). With AO, the PSF became sharper and brighter with the Strehl ratio enhancement of 1.93 (bottom right in Fig. 2c). This improvement in PSF led to 1.62-fold localization number increase and localization precision enhancement from $\sigma_{AO\ off} = 52$ nm to $\sigma_{AO\ on} = 34$ nm in SMLM imaging (Fig. 4d, e). Without AO, the cilia were only vaguely visible with reduced contrast and resolution. However, AO clearly resolved the hollow cylindrical

structures of the ciliary membranes with a spacing measured as 180 nm (Fig. 2e).

Next, we imaged oligodendrocytes in whole *Tg(clau-dinK:gal4vp16;uas:megfp)* zebrafish larvae with GFP expressed on oligodendrocyte membranes. This line was selected because it is a widely-used animal model for investigating myelination of the central nervous system in vertebrates and neurodegenerative diseases such as amyotrophic lateral sclerosis (ALS) and Huntington's disease[32]. For SMLM

**Fig. 4 | Deep-tissue SMLM images of whole zebrafish larvae without and with AO. a** A diagram showing a whole zebrafish larva mounted on a cover glass. **b** Confocal fluorescence images of the whole body (left column) and hindbrain (right column). Dashed square indicates hindbrain region. Colored squares roughly indicate SMLM imaging regions. Scale bars indicate 500 μm (left column) and 50 μm (right column). **c–e** SMLM imaging of cilia in a whole 3-dpf zebrafish at the depth of 82 μm. Aberration correction map (top row in **c**), ensemble-averaged normalized PSFs of the first 10,000 frames (bottom row in **c**), and SMLM images (**d**). Magnified views of regions indicated by arrows in (**d**) (left column in **e**) and cross-sectional profiles of white boxes (right column in **e**). Color bars in (**d**, **e**)

indicate localization numbers. Scale bars indicate 500 nm (**c**), 5 μm (**d**), and 500 nm (**e**). **f–h** Same as (**c–e**), but for oligodendrocytes at the hindbrain in a whole 5-dpf zebrafish at the depth of 100 μm. Scale bars indicate 500 nm (**f**), 2.5 μm (**g**), and 500 nm (**h**). **i–k** Same as (**f–h**), but for oligodendrocytes near spinal cords in a whole 3.5-dpf zebrafish at the depth of 52 μm. Scale bars indicate 500 nm (**i**), 2.5 μm (**j**), and 500 nm (**k**). **l**, **m** Same as (**i**, **j**), but for oligodendrocytes near spinal cords in a whole 5-dpf zebrafish at the depth of 102 μm. Arrowheads indicate sites with FWHM measured as 160, 150, and 160 nm from bottom to top. Scale bars indicate 500 nm (**l**) and 5 μm (**m**). Source data are provided as a Source Data file.

imaging, oligodendrocytes were immunolabeled with anti-GFP antibodies conjugated with Alexa Fluor 647 after fixation (see "Preparation of zebrafish" and "Immunolabelling of commissural tracts at hindbrain of zebrafish larvae" sections in "Methods"). At the depth of 100 μm in a whole 5-dpf zebrafish, we obtained images of commissural tracts at the caudal hindbrain (Fig. 4f–h). Commissural tracts[33] are thin, ladder-like structures perpendicularly crossing a Y-shaped, thick branches of spinal cords (right column in Fig. 4b). The RMS wavefront distortion of the aberration (top row in Fig. 4f) was 3.08 rad, which was even higher than that of artificial aberration layer in Fig. 2. Although this high-order aberration caused PSF blur (bottom left in Fig. 4f), AO restored PSF successfully, yielding localization precision improvement from $\sigma_{AO\,off} = 67$ nm to $\sigma_{AO\,on} = 38$ nm. Consequently, oligodendrocytes wrapping thin axons were clearly resolved with AO (Fig. 4g, h). The FWHMs of the cross-sectional profiles of multi-layered oligodendrocyte membranes in myelin sheath were measured as 240 nm (left-hand peak) and 220 nm (righthand peak) with a spacing of 330 nm (Fig. 4h).

We also conducted imaging of oligodendrocytes near spinal cords. At a depth of 52 μm in a 3.5-dpf zebrafish larva, we captured early myelination (Fig. 4i–k). The RMS wavefront distortion of the aberration (top row in Fig. 4i) was 2.13 rad. AO recovered the PSF blur resulting from this aberration, yielding Strehl ratio enhancement of 3.34, 8.59-fold localization number increase, and localization precision improvement from $\sigma_{AO\,off} = 53$ nm to $\sigma_{AO\,on} = 35$ nm. In virtue of this improvement, many invisible structures became visible with AO (Fig. 4j), accompanying localization number enhancement up to 37.4 fold. Especially, nascent myelin sheaths[34] formed by oligodendrocytes at the early myelination step were clearly captured with AO (arrows and arrowheads in Fig. 4j). The FWHMs of their necks were measured as 120 nm (arrowhead in Fig. 4j) and 140 nm (arrowhead in Fig. 4k).

Lastly, we conducted imaging of an older zebrafish with complete myelination. At a depth of 102 μm near spinal cords in a whole 5-dpf zebrafish, we observed oligodendrocytes wrapping a Mauthner axon[35] (Fig. 4l, m). The RMS wavefront distortion of the aberration (top row in Fig. 4l) was 2.64 rad. This aberration resulted in highly elongated single-molecule PSFs along the axon (bottom left in Fig. 4l). The tissue aberration was so severe that localization processes almost failed, making the entire structures invisible (Fig. 4m). However, AO restored near-ideal PSF, yielding Strehl ratio enhancement of 8.86 (bottom right in Fig. 4l). This caused 16.2-fold localization number increase and localization precision of $\sigma_{AO\,on} = 34$ nm. Here, $\sigma_{AO\,off}$ was not measurable due to lack of localization number. Meanwhile, the oligodendrocyte membranes wrapping a Mauthner axon was clearly visible with AO, including thin axons with FWHMs measured as 150–160 nm (arrowheads in Fig. 4m).

## Discussion

This work introduced an AO-SMLM modality for investigating highly aberrated specimens such as thick tissue slices and whole zebrafish. CLASS microscopy was employed as a label-free wavefront-sensing AO method to determine tissue aberration based on intrinsic reflection signals. Then, its corresponding correction map was applied to an SLM in the fluorescence emission path of the SMLM setup to reconstruct

super-resolved fluorescence images. Our method increases the degree of correctable aberration by more than three times in terms of the RMS wavefront distortion compared with previous AO-SMLM methods due to the use of intrinsic reflectance, unique linear optimization algorithm finding the tissue aberration, and the physical correction of aberration with a liquid-crystal SLM (see "Comparison of CLASS-SMLM with the current state-of-the-art AO-SMLM" section in Supplementary Note 2 for a detailed comparison between our method and previous AO-SMLMs). Essentially, our AO-SMLM corrects unexplored degrees of aberration and achieves SMLM image quality similar to those of samples with little aberration.

Our label-free AO with a strong aberration correction capability enabled AO-SMLM imaging of whole zebrafish larvae. Though oblique-plane SMLM without AO was applied to the tail fins of whole stickleback fish[36], the imaging depth was limited to 44 μm through the ~33-μm-thick caudal fin epidermis. Also, AO-SMLM has not been applied to whole animals probably due to multiple challenges such as high scattering and aberration of the tissue, high background fluorescence, low photon output of fluorescent proteins, limited labeling depth of immunostaining, etc. We circumvented the scattering issue by choosing transparent zebrafish and suppressed the fluorescence background by choosing localized structures (e.g., cilia). Also, we adapted the whole-mount immunolabeling protocol with additional steps for lipid extraction and protein digestion for the representative SMLM dye, Alexa Flour 647. CLASS combined with these efforts has enabled us to obtain SMLM images from various areas of the nervous system of whole zebrafish from the head (e.g., hindbrain) to the back (e.g., spinal cord).

Comparison of both AO-off and -on SMLM images elucidates the major issues in deep-tissue SMLM imaging of highly aberrated samples. In most cases, a substantial loss of localization number (up to 97.3%; arrowhead in Fig. 4j) was a major issue. The localization precision was also degraded, yet the difference was relatively small (up to 3.61 times; Fig. 2g, h). Our AO-SMLM resolved these problems by restoring distorted single-molecule PSFs even up to 3.08 rad of RMS wavefront distortion (Fig. 4g). It led to a localization number increase by up to 37.4 times and localization precisions improvement from 52–67 nm to 34–38 nm at a depth of up to 102 μm. In contrast, the previous AO-SMLM methods that iteratively optimized image quality metrics imposed 1 rad limit of RMS wavefront distortion. This limit could have been set by the weak single-molecule signals overwhelmed by the background that made it impossible to evaluate image quality metrics. These methods could enhance the localization number only by ~2–8 folds at a depth of up to ~50 μm[13,18]. We found that the number of control elements used in the SLM plays an important role in determining the complexity of aberration that can be corrected and the width and brightness of the recovered PSFs (see "Aberration correction maps with different spatial frequency resolution" section in Supplementary Note 2).

CLASS identifies aberration without using fluorescence. That is, photobleaching does not occur during the aberration calculation. Therefore, our AO-SMLM has a crucial advantage when the labeling efficiency is low or multicolor SMLM is required. For example, the previously presented approach that uses two-photon fluorescence

spots as nonlinear guide stars requires additional labeling and, thus, the allocation of additional fluorescence channels for guide stars[16]. This may not be easy in multicolor SMLM. For example, several different emission wavelengths are used for SMLM imaging of synapses[37,38]. In this case, adding an additional channel may be expensive or impossible. Of course, single-molecule images of SMLM fluorophores can be used for finding aberrations. However, due to their weak intensities, a significant portion of single-molecule emission PSFs must be used for aberration estimation, which reduces the density-limited resolution of the resultant SMLM image.

Further increase in imaging depth may require suppression of background fluorescence (Supplementary Fig. 6). One option is to combine our system with selective illumination methods such as two-photon activation[12], selective-plane illumination[39], or oblique-illumination/detection schemes[36]. One may consider employing tissue clearing to reduce scattering, but there still exists the need for correcting aberrations (see "Aberration of a tissue-cleared whole zebra-fish" section in Supplementary Note 2). Taking into account the importance of brain and zebrafish imaging in studying neurodegenerative diseases, gene functions and development of vertebrates including humans, our study will enable the sub-diffraction investigation of vertebrate-specific topics in genetics, developmental biology, and neurobiology at a whole organism level.

## Methods

### Ethical statement
All experimental procedures were approved by the Committee of Animal Research Policy of Korea University (approval numbers KUIACUC-2019-24 and KOREA-2021-0037).

### CLASS microscope setup
A low-coherence 678 nm laser (SLD-261-HP2-DBUT-PM-PD-FC/APC, Superlum, coherence length: ~40 μm) was used to illuminate the samples for time gating. The beam was steered by a two-axis galvo mirror [6210H, Cambridge Technology; controlled with a custom-developed code (MATLAB2020a, MathWorks)] to scan the illumination angle. It was then divided into the sample beam (SB) and reference beam (RB). The RB passed through a diffraction grating (Ronchi 120 lp/mm, Edmund Optics) for off-axis interference imaging. Only the first-order diffraction of the RB was combined with the SB reflected from the sample at the beam splitter in front of the sCMOS camera [pco.edge 4.2 m, PCO; controlled with dedicated software (Camware 4.05, PCO)]. We recorded interference images while scanning the illumination angles in such a way to cover the entire numerical aperture (1.2) range of the objective lens. Typically, we recorded 4000 images to uniformly cover the pupil plane of the objective lens. The imaging depth was selected according to the objective lens focus and the position of a reference mirror mounted on a motorized actuator [Z825B, Thorlabs; controlled with dedicated software (APT Version 3.21.6, Thorlabs)]. The complete CLASS setup was built on a commercial inverted microscope (Eclipse Ti2-E, Nikon) equipped with a 60×/1.2 NA water immersion objective lens (UPLSAPO 60XW, Olympus). In addition, a reflection matrix was constructed based on the measured complex field maps, and the CLASS algorithm was applied to obtain the sample-induced aberrations (see "Detailed process of CLASS algorithm" section in Supplementary Note 1 for the detailed process). The recording of complex field maps takes 35 s, and CLASS calculation takes less than 7 s for an area of ~33 × 33 μm², the FOV in the present study. Once aberration is evaluated, it takes less than a second to generate its correction map and display it on the SLM. Therefore, the total time for 2285 orthogonal angular modes is 43 s, which translates to only 18.8 ms per angular mode. Compared to other AO-SMLM methods, CLASS-SMLM is much faster by one or two orders of magnitude in terms of correction speed per mode.

### SMLM setup
GFP or Alexa Fluor 647 were excited with a 488 nm laser (OBIS 488-60 LS, Coherent) or a 647 nm laser (2RU-VFL-P-2000-647-B1R, MPB Communications Inc.), respectively. A 405 nm laser (405 nm LX 50 mW laser system, OBIS) was used for the activation of Alexa Fluor 647. These lasers were coupled and focused onto the objective back aperture for epi-illumination. A 505 nm LED (M505L3, Thorlabs) was installed for low-magnification transmission imaging. It was used to select the region of interest. LED transmission images were acquired using a camera [LM135M, Teledyne Lumenera; controlled with dedicated software (Lucam Software v6.8.3, Teledyne Lumenera)]. A motorized filter wheel [FW102C, Thorlabs; controlled with dedicated software (FWxC 5.0.0, Thorlabs)] was used for controlling the intensity of excitation beam. The fluorescence emitted from the sample was magnified by a factor of 2 to achieve a pupil diameter of 9.5 mm at the SLM [X13138-06, Hamamatsu; controlled with a custom-developed code (MATLAB2020a, MathWorks)]. A set of 4f relays was designed for the final magnification (×100), which yielded a 130 nm effective pixel size at the EMCCD camera [DU-888U3-CS0-#BV, Andor; controlled with NIS software (NIS-Elements AR 5.30.03 64-Bit, Nikon)]. Emission filters (ET700/75 m and ET525/50 m, Chroma) were placed in front of the EMCCD camera to filter out unwanted signals. The SMLM setup and CLASS microscope shared the objective and tube lenses (see Supplementary Fig. 7 for full experimental setup). The acquisition time for SMLM imaging depends on the sample type. Usually, the cell monolayer required the shortest time of ~30 min (50000 frames) because there was almost no scattering-induced photon loss. When imaging thick brain slices or whole zebrafish, a larger number of image frames were required due to the localization loss by the scattering-induced photon loss and background noise.

### SMLM analysis
All raw images of the blinking single-molecule emission PSFs were processed with the ThunderSTORM ImageJ plugin (imageJ version 1.53t, National Institutes of Health; ThunderSTORM 1.3, GitHub) to find their centroids by the fitting[40]. The localization precision values of the obtained SMLM images were calculated using the nearest neighbor analysis. It was assumed that a localized position $\mathbf{r}_i(t_i)$ in a frame at time $t_i$ has a nearest neighbor localized position $\mathbf{r}_{i,NN}(t_{i+1})$ in the next frame at time $t_{i+1}$. The pairwise displacement $d_i$ is defined as the displacement $d_i \equiv \mathbf{r}_i(t_i) - \mathbf{r}_{i,NN}(t_{i+1})$ between nearest neighbors. Once collected, the set of pairwise displacements $d_i$ can be presented in the form of a histogram. Subsequently, the envelope of this histogram is fitted with a non-Gaussian function with a Gaussian correction term. The correction term is required because some nearest neighbors may result from different molecules in proximity rather than from identical molecules. Therefore, the fitting curve is expressed as follows:

$$p(d) = A_1 \frac{d}{2\sigma^2} \exp\left[-\frac{d^2}{4\sigma^2}\right] + A_2 \frac{1}{(2\pi\omega^2)^{1/2}} \exp\left[-\frac{(d-d_C)^2}{2\omega^2}\right],$$

where $d$ is the pairwise displacement, $\sigma$ the Gaussian standard deviation defining the localization precision, and $\omega$ the Gaussian standard deviation of the correction term centered at $d_C$.

### Comparison of CLASS-SMLM with previous AO-SMLM
As discussed in "Comparison of CLASS-SMLM with the current state-of-the-art AO-SMLM" section in Supplementary Note 2, we compared CLASS-SMLM with REALM[18]. The authors provided a free open-source code for REALM. However, it was not compatible with our SLM, so we rewrote REALM code in MATLAB language (MATLAB2020a, MathWorks).

## Preparation of nanoparticle-coated cover glasses

In total, 18 × 18 mm² cover glasses (0101030, Marienfeld) were sonicated in 1 M KOH solution (6592-3705, Daejung) for 30 min and then washed with Milli-Q water (Direct 8, Merck) to remove remaining KOH solution. When necessary, the sonicated cover glasses were coated with gold nanoparticles. In this case, the cover glasses were previously coated with poly-L-lysine (P8920-100ML, Sigma Life Science) for 5 min. Subsequently, 5× diluted 100 nm diameter gold nanoparticles (753688-25 ML, Sigma-Aldrich) were dropped onto the cover glasses. They were dried in an oven at -60 °C overnight.

## Immunolabelling of microtubules in COS-7 cells

The COS-7 cells [AC28806, Korean Collection for Type Cultures (KCTC)] were seeded on gold-nanoparticle-coated cover glasses and cultured overnight in DMEM (11965-092, Gibco) containing 10% FBS (10082147, Thermo Fisher) and 1% penicillin-streptomycin (15070063, Thermo Fisher). Prior to fixation, the cells were washed three times with pre-warmed PBS (37 °C; 21-040-CV, Corning) and treated with the pre-warmed extraction buffer (37 °C; 0.125% Triton X-100 (X100-500ML, Sigma-Aldrich) and 0.4% glutaraldehyde [G7526-10ML, Sigma-Aldrich) in PBS]. Immediately after the treatment, the extraction buffer was quickly aspirated, and the cells were rinsed three times with PBS. Next, the cells were fixed with the pre-warmed fixation buffer (37 °C; 3.2% paraformaldehyde (PC2031-100-00, Biosesang) and 0.1% glutaraldehyde in PBS for 10 min at room temperature (RT). After fixation, fixatives were quenched with 10 mM fresh sodium borohydride in PBS for 5 min at RT while being shaken. After three washing cycles with PBS, the cells were permeabilised with 3% BSA (A7030-50G, Sigma Life Science) and 0.5% Triton X-100 in PBS. Then, the primary antibody for tubulin (ab6046, Abcam), which was previously diluted 1000× in the blocking buffer (3% BSA and 0.5% Triton X-100 in PBS), was added to the cells for 1 h at RT on a rocking platform. The cells were then washed three times with PBS and treated with the secondary antibody conjugated with Alexa Fluor 647 (A-21245, Thermo Fisher), which was 1000× diluted in the blocking buffer for 1 h at RT, while being shaken. After immunolabelling, the cells were washed three times with PBS and stored at 4 °C before the AO-SMLM imaging.

## Preparation of mouse brain slices

All experimental procedures were approved by the Committee of Animal Research Policy of Korea University (approval number KUIA-CUC-2019-24). Adult (over 8 weeks old) Thy1-EGFP line M (Jackson Labs #007788) mice were deeply anesthetised with an intraperitoneal injection of 100 mg/kg ketamine and 10 mg/kg xylazine. After decapitation, their brains were quickly excised and dropped into an ice-cold artificial cerebrospinal fluid (ACSF). The brains were cut into 150–200 μm thick coronal slices [typical dimension: ~10 mm × 8 mm × (150–200 μm)] with a vibroslicer (World Precision Instruments, Sarasota, FL, USA) and fixed at 4 °C in 4% paraformaldehyde overnight before immunolabelling.

## Preparation of zebrafish

All experimental procedures were approved by the Committee of Animal Research Policy of Korea University (approval number KOREA-2021-0037). *Tg(bactin2::Arl13b-GFP)* (#FRZCC1009) and *Tg(claudinK:gal4vp16;uas:megfp)* (#FRZCC1013) zebrafish embryos (typical dimension: ~2.5 mm × 500 μm × 500 μm) were raised at 28 °C in E3 embryo medium [5 mM NaCl (7548-4405, Daejung), 0.17 mM KCl (6566-4400, Daejung), 0.33 mM CaCl₂ (2507-1400, Daejung), and 0.33 mM MgSO₄ (21032, Daejung)]. After hatching, they were transferred to E3 medium containing N-phenylthiourea (P7629, Sigma-Aldrich) to inhibit pigmentation. After 3–5 days, the larvae were anesthetised with tricaine (E10521, Sigma-Aldrich) in E3 medium and fixed in 4% paraformaldehyde for 1–2 h at RT.

## Immunolabelling of mouse brain slices and zebrafish

The fixed brain slices and whole zebrafish larvae were permeabilised for 3 h with a blocking buffer (3% BSA and 0.5% Triton X-100 in PBS). Subsequently, the slices were immunolabelled for 5 days with 200× diluted anti-GFP Alexa 647-conjugated primary antibodies (A31852, Thermo Fisher) in the blocking buffer. Zebrafish larvae were immunolabelled in the same way but overnight.

## Immunolabelling of commissural tracts at hindbrain of zebrafish larvae

Hindbrain is hardly immunolabelled via standard protocols because they are located deep beyond the surface at a depth ranging 90–120 μm for 3-dpf zebrafish larvae. Therefore, for immunolabeling commissural tracts, additional treatments are required for lipid extraction and protein digestion. To this end, fixed zebrafish larvae firstly treated with acetone (650501, Sigma-Aldrich) at −20 °C for 7 min. Then, they were treated with 0.25% trypsin (SH30042.01, Cytiva) at ice for 15 min, followed by treatment with 1 mg/ml collagenase (C9891-100MG, Sigma-Aldrich) at RT for 40 min. The next step was 30-min treatment with 10000× diluted 10 mg/ml proteinase K (P4850-1ML, Sigma-Aldrich) at RT. Subsequently, the zebrafish larvae were permeabilized for 1 h at RT with a blocking buffer [5% BSA (A9647-10G, Sigma-Aldrich), 5% sheep serum (013-000-1210, Jackson ImmunoResearch) diluted in PBS (21-040-CV, Corning) containing 0.1% Triton X-100 (X100-100ML, Sigma-Aldrich)]. The final step was incubation in anti-GFP Alexa 647-conjugated primary antibodies (A31852, Thermo Fisher) diluted 200× in the blocking buffer overnight.

## Imaging buffer preparation

Imaging buffer base [10% (m/v) glucose (G7021-100G, Sigma-Aldrich), 50 mM Tris-HCl, pH 8.0 (TR2016-050-80, Biosesang), and 10 mM NaCl], dilution buffer (50 mM NaCl and 10 mM Tris-HCl, pH 8.0) and GLOX [14 mg glucose oxidase (G2133-250KU, Sigma-Aldrich), 50 μl of catalase (C3155-50MG, Sigma-Aldrich), and 200 μl of dilution buffer] were prepared in advance. Then, right before starting experiment, imaging buffer was freshly prepared by 100× diluting GLOX and 1% βME (M3148-25ML, Sigma-Aldrich) in imaging buffer base. Every sample was soaked in imaging buffer for inducing photoswitching of Alexa 647 dyes.

## Tissue clearing of zebrafish embryos

Tissue clearing was performed using a tissue clearing kit (HRTC-001, Binaree). Fixed *Tg(claudinK:gal4vp16;uas:megfp)* zebrafish embryos were immersed in the starting solution of the kit at 4 °C. Then, they were immersed in the tissue clearing solution A of the kit at 37 °C for 1 h. After washing them with Milli-Q water for 30 min four times, they were immersed in the tissue clearing solution B of the kit at 37 °C for 1 h. After that, they were washed with Milli-Q water for 30 min four times. Finally, they were immersed in the mounting & storage solution (SHMS-060, Binaree) and incubated at 37 °C for overnight.

## Statistics and reproducibility

No statistical method was used to predetermine the sample size. Because of the heterogeneity of biological samples, all kinds of samples were imaged many times. Then, only parts of them were selected. In detail, for drawing figures, the best results were chosen among 52 (Figs. 1c, d and 2a–d), 48 (Fig. 3a, f and Supplementary Fig. 5a–e), 7 (Fig. 4b), 128 (Fig. 4d), 64 (Fig. 4g), 503 (Fig. 4j, m and Supplementary Fig. 5f–h), 10 (Supplementary Fig. 3a–d), 9 (Supplementary Fig. 10), 5 (Supplementary Fig. 11), 5 (Supplementary Fig. 15), 28 (Supplementary Fig. 16), 14 (Supplementary Fig. 17), and 3 (Supplementary Fig. 18) different images. Excluded images were

discarded because they were accidentally defocused or there are no interesting structures.

## Reporting summary

Further information on research design is available in the Nature Portfolio Reporting Summary linked to this article.

## Data availability

Source data are available from https://doi.org/10.5281/zenodo.8080248. Source data are provided with this paper.

## Code availability

The code used for aberration calculation from the recorded reflection matrix was previously published[21]. All raw images of the blinking single molecules were processed with the ThunderSTORM ImageJ plugin (imageJ version 1.53t, National Institutes of Health; ThunderSTORM 1.3, GitHub)[40]. Custom scripts written in MATLAB (MATLAB2020a, Math-Works) were used to control the experiment and analyze the data: the codes for displaying a correction map on the SLM, measuring the width of subcellular structures, calculating important parameters such as localization precision. They are all available from https://doi.org/10.5281/zenodo.8080248.

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

## Acknowledgements

This work has been financially supported by the Institute for Basic Science (IBS-R023-D1) and the National Research Foundation (NRF) grants funded by the Korean government (MSIT) [R1612208 (S.P., Y.J., J.H.H., and W.C.), 2021R1A2C2010792 (M.K., S.K., and S.-H.S.), 2021R1A4A1032114 (M.K., S.K., and S.-H.S.), 2021M3H9A1097594 (S.K.* and H.C.P.), 2019R1I1A1A01059015 (S.P.*), and 2022R1A6A3A01085960 (S.P.*)]. FRZCC #1009 and 1013 were provided by the Fluorescent Reporter Zebrafish Cooperation Center (FRZCC), Republic of Korea. S.P. and S.P.* indicate Sanghyeon Park and Sangjun Park, respectively. S.K. and S.K.* correspond to Sangyoon Ko and Suhyun Kim, respectively.

## Author contributions

S.-H.S. and W.C. conceived the project. S.P. and Y.J. constructed the experimental setup. S.P. conducted super-resolution imaging experiments with the help of M.K., S.K. and S.P.* With guidance of Y.J., J.H.H., S.-H.S. and W.C., S.P. performed data analysis. J.H.H., S.K.* and H.C.P. prepared biological samples and provided guidance for data interpretation. S.P., S.-H.S. and W.C. prepared the manuscript and all authors contributed to finalizing the manuscript. S.-H.S. and W.C. supervised the project. S.P. and S.P.* indicate Sanghyeon Park and Sangjun Park, respectively. S.K. and S.K.* correspond to Sangyoon Ko and Suhyun Kim, respectively.

## Competing interests

The authors declare no competing interests.
