## [Peer Review File · Nature Communications]

REVIEWER COMMENTS

Reviewer #1 (Remarks to the Author):

Review report on “Label-free adaptive optics single-molecule localization microscopy for whole animals” by Sanghyeon Park et. al., where the review is particularly focused on optical methods used in manuscript such as implementation of the wavefront sensing and details relating to the spatial light modulation presented in the manuscript.

This work reports the application of label-free wavefront sensing adaptive optics (AO) to single-molecule localization microscopy (SMLM) for deep-tissue super-resolution imaging. A number of investigations on highly aberrated specimens such as thick tissue slices and intact organisms with and without AO were presented and compared together. A closed-loop accumulation of single scattering (CLASS) microscopy was employed as a label-free wavefront-sensing AO method to determine the sample aberration based on intrinsic reflection signals. With the aid of AO-SMLM the authors were able to image thick (thickness of 150-200 μm) brain slices of Thy1-EGFP transgenic mice, where without AO system the SMLM imaging has suffered from shallow imaging depth of sub-100 μm thickness. The used methods for the imaging, wavefront sensing, and aberration removing are clearly presented in the manuscript or in the Supplementary Information. In my opinion results deserve publication. However, I have some comments on the manuscript and Supplementary Information. My comments are mostly focused on the implementation of the wavefront sensing and the optical methods used in the study, and hope to receive proper response before acceptance of the work.

My comments:

SMLM is basically based on the fact that the spatial coordinates of single fluorescent molecules (also called emitters) can be determined with high precision if corresponding PSFs do not overlap. Subpixel shifts in the coordinates of an emitter lead to predictable changes in pixel intensities that can be used to compute its precise location. The localization precision reflects the scatter of localizations that would be obtained if a molecule was imaged and localized many times, and is basically limited by the signal to noise ratio (SNR) and not by the wavelength of light or the pixel size. To avoid overlaps between the PSFs of individual molecules, fluorescent emissions of distinct molecules are separated in time; the most common approach to obtain this temporal separation uses the phenomenon of photoswitching, where fluorescent molecules can switch between an active ‘ON’ (also called ‘bright’) state, where they emit fluorescent light when excited, and one or more inactive ‘OFF’ (also called ‘dark’) state in which they do

not fluoresce. In the current work a label-free wavefront sensing adaptive optics method is applied on SMLM for deep-tissue super-resolution imaging. The details of processes used for improving PSFs of single molecules and differences with the conventional SMLM which is reviewed above should be emphasized. Actually, in the conventional SMLM, location of a single fluorescent molecule when adjacent molecules are 'OFF' can be precisely determined by determining the maximum intensity of the corresponding PSF even without using the PSF distribution. In the proposed AO-SMLM, AO helps to have a narrower PSF for a given single molecule when the other molecules' PSFs do not overlapped. During whole area imaging of a sample by AO-SMLM, what arrangements were considered to prevent such overlapping of PSFs. Please clarify. Could you add some details about your SMLM. Do you use photoswitching? Do you use any fitting on the individual molecule PSF for more precise determining its location? In the use of CLASS, still the resolution is limited by the diffraction limit. Is the super-resolution imaging advantage of the method only due to the use of AO-SMLM? It is necessary to emphasize that the restoration of abnormal PSFs to near-ideal PSFs of adjacent molecules are done at the different time? Or am I wrong?

In the current work, collected reflectance images from biological structures are used for evaluating the phase aberration induced by the tissue on the wavefront via CLASS microscopy, and the derived phase aberrations are successfully applied for imaging thick samples via AO-SMLM microscopy. Since phase aberration measurement from the reflectance images is one of the key parts of the current work, in my opinion it deserves and in some places it is needed to add details of the phase aberration deriving from the collected reflectance images. My main concerns are in this regard, which have been expressed below under comments on Supplementary Information, especially regarding Supplementary Figure SN7.

It would be useful to determine and add some additional details about phase extraction method, such as the time is needed for a given phase extraction, and the time is needed to apply the necessary corrections in order to remove aberrations from the sample wavefront. How long does it take to capture an AO-SMLM image? Is it possible to apply the used method for imaging of a sample having dynamic feature?

In most of the experimental figures there are some images of the samples having non-equal dimensions in the x and y directions, while the aberration correction maps were presented in circular frames. In which field of view the phase correction is applied? Is it cover all imaging area? Please add some clarification.

In which range of angles the tissue structures were illuminated? What is the total number of images are used for a given aberration correction. Can you add details about applying different illumination angles?

In lines 161 and 162 it was mentioned that “Considering the microtubule diameter (approximately 25 nm) and size of primary and secondary antibodies (10-15 nm), this value agrees with well-known widths of microtubules.” Are all microtubules the same size?

What is the number of wavefront cells of AO correction? How it affect restored PSF?

A definition for “localization number” is needed.

It would be nice to introduce the signal to noise ratio (SNR) at the first use.

In lines 176 and 177 change “A SMLM images of ...” to “a SMLM images of ...”.

Please add details about plots Figs. 2i-2l, Fig. 3j and Fig. 3j,k, or add some introduction to the nearest neighbor analysis.

My comments on Supplementary Information:

In line 152, change “The 678-nm laser beam” to “The 678 nm laser beam”.

In supplementary Not, lines 160 – 163 you mentioned that “After being reflected from the reference mirror, RB retraced the same path back and traveled toward the grating through BS3. Passing through the grating, RB was diffracted into several branches. Among them, only the first-order branch was selected for oblique incidence of RB at the camera plane. The other branches were discarded by blocking with an iris.” and as is shown in Supplementary Figure SN1, you used a grating after BS3. In my opinion you can remove the grating and iris and only by a small rotation of the mirror used in the path, again you have the desired angle between RB and SB on the sCMOS camera. Please mention the reason for the use of grating?

In the same figure, Supplementary Figure SN1, a mirror is missing after BS2 in the light return path.

In Supplementary Not, lines 164 – 165 you mentioned that “At the camera plane, SB and RB formed an interference pattern when the length difference between them was shorter than the coherence length.” There is no any additional information about SB reflected from the sample plane. Is it focused on a given

point on the sample or it covers whole area of the sample? What is the size of area illuminated by the SB? How the aberrations of the sample affect and deviate the SB? Is SB still an almost plane wave with small phase fluctuations, after reflecting from the sample. Please clarify these. You have mentioned previously that the sample was illuminated widefield, but it is not clear that the phase aberrations of the sample how modulates entire SB wavefront? What is the effect of depth in the sample on the aberrations, and how can consider the aberrations of different depths in the tissue. Do you imaged the reflected SB on the sCOMS camera? Do you have considered the specimen includes a transparence shield and a main structure, in which the sample beam is reflected from the main structure and double-passed through the transparent shield (which is mainly a non-homogenous and non-flat layer)? Could you explain the origin of phase aberrations implemented on the reflected beam with more details?

In lines 166-171, it is necessary to clarify the purpose for the point illumination of the sample and imaging of it on sCMOS through OL2? In line 170, is it ok still you call the reflected beam as RB?

In Supplementary Figure SN7a and especially according to its Fourier transform in Supplementary Figure SN7b, a linear fringes interference pattern (at least over the bright area) should be appear? Why such pattern does not appear? Can you show an enlarged pattern to present such interference fringes, even in an inset in Supplementary Figure SN7a? What was the sample? Is it a reflectance phase or amplitude test plate? Did you record the interference pattern with the aid of reflected sample beam? Could you determine the size of sample under study and size of area imaged by the system?

You scanned incident angle of the beam on the sample surface and record different interference patterns, what is the value of scanning angle step? How affect the value of scanning angle step on the resolution of phase aberration extraction?

In Supplementary Figure SN7d, you mentioned that the color bar shows amplitude of the inverse Fourier transform of selected spectral component in c. Since you are interested to determine the phase aberration of the sample, it is a bit confusing. Please clarify. I think that you presented an infinite mode interference pattern of the RB and SB in Supplementary Figure SN7d, where there is no angle between the interfering beams (I can address to such interference pattern in [S. Rasouli, F. Sakha, and M. Yeganeh, "Infinite-mode double-grating interferometer for investigating thermal-lens-acting fluid dynamics," Meas. Sci. Technol. 29, 085201 (1-10), 2018]). It seems that infinite mode interference patterns can be used in CLASS and omit one step from phase aberration process such as Supplementary Figure SN7b. In this case other phase extracting methods should be used.

In lines 382 and 383 you mentioned that "At this moment, aberration map is obtained by reshaping phase array applied to the rows of $R(k_{out}, k_{in})$ into a square form (Supplementary Figure SN7j)." I am interested in to know how you establish a relation between different k_{in} or the corresponding illumination angle and different position on the sample area?

Reviewer #2 (Remarks to the Author):

The paper presents a method for single molecule localization microscopy that attempts to correct the aberrations induced by tissue scattering. Results are shown on both brain tissue slices and whole zebrafish. The results indicate a modest improvement in resolution and ability to image deeper in the presence of tissue scattering.

Novelty: The main concern/limitation of the work as presented is that it is not clear what is new/novel about their technique. Essentially the paper completely relies on CLASS (a very elegant and interesting past work) to estimate aberrations in moderately scattering specimens -- and then just uses phase conjugation to correct for these measured aberrations. The only reason for CLASS as a technique to be interesting is to correct for aberrations -- and even in their earlier papers they already demonstrate that. The only new part seems to be that in this paper they use it for single molecule localization microscopy -- but that's straightforward in the cases where CLASS works.

Experiments: Experimental results indicate a slight improvement quantitatively --- about a 2-3x improvement in resolution compared to non-AO which indicates that the level of scattering in the samples are quite small to begin with. The authors suggest that previous SMLM techniques operate with wavefront aberrations of the order of 1 radian and suggest that their proposed technique can handle much larger aberrations but their bio-samples are not all that much more scattering. The brain sample has a RMS wavefront distortion of ~ 1 , while the zebrafish it is ~ 2 . Furthermore, the quantitative comparisons seem to be limited to the insets rather than averaged over the entire sample. The insets in all experiments seem to have been chosen to highlight the difference between nonAO and their method (which is understandable and what one would expect of authors). But one would expect that quantitative metrics of comparison such as resolution enhancement etc are averaged over the entire sample rather than be calculated and presented only on these author-selected regions. This makes it difficult to evaluate what the actual improvements in performance are.

Claims: In many places, the paper overclaims its contributions vis-a-vis the state of art. The "whole animals" in the title should be replaced with zebrafish or otherwise toned down. After all one particular reason the zebrafish is such a well studied model organism for optical microscopy techniques is because it has so little scattering compared to most other animal tissue.

Lack of comparisons: As the paper readily points out in their introduction, there have been several techniques that have been developed for AO in single molecule localization microscopy. But the current manuscript does not directly compare with any of them at all. Instead the quantitative comparisons are

only with baseline SMLM without any adaptive optics, which is really not a fair baseline. In order for the paper to demonstrate an improvement over current state of art, direct comparisons with current state of art AO methods in SMLM need to be shown. Just a qualitative claim that current methods have not previously handled aberrations greater than 1 rad is not sufficient at all.

Reviewer #3 (Remarks to the Author):

In the present manuscript, Park et al applied label-free wavefront sensing adaptive optics to SMLM (single-molecule localization microscopy) for deep-tissue super-resolution imaging. The authors succeeded in resolving sub-diffraction morphologies of cilia and oligodendrocytes in whole mount zebrafish as well as dendritic spine in thick mouse brain tissues at a considerable depth (around 100 micro-meter). The authors claim that the approach can expand the application range of SMLM to intact animals that cause the loss of localization points owing to severe tissue aberrations.

Major comments:

1) While it is a good idea to apply label-free CLASS to AO, an in vivo imaging with CLASS and an AO-scanning microscopy has already been reported by the same authors (Nature Communications, 2019). In this sense, I don't think there is a satisfactory conceptual leap. With this reason, I am not convinced that the manuscript is enough for Nature Communications. I feel that the manuscript would be suitable for publication in a more specialized journal.

2) The aberration correction maps shown in Figures 1, 3, and 4 are inferior to those presented in the previous studies (Figure 4, Nature Communications 2017; Figure 2, Nature Communications 2019). This is presumably because the corrections were made in a large field. If this is the case, the authors need to state this clearly.

3) The authors applied the technique to fixed samples, not live animals. For fixed samples, tissue-clearing (with Scale, Clarity, SeeDB, Cubic, etc) is much more convenient and powerful than AO, and thus, the authors first need to employ tissue-clearing. After doing so, the authors need to evaluate the power of the method described in the present manuscript. Otherwise, the results cannot be evaluated properly in a practical sense. Also, for the evaluation, the authors need to compare the images of AO with images obtained by a commercially available high-resolution microscope such as Zeiss AiryScan.

4) Fixed samples cannot be called “intact animals”. If the authors want to use the term “intact animals”, samples need to be alive. Upon applying the technique to live intact animals, it would be critical how much time is needed for the correction.

Reviewer #4 (Remarks to the Author):

Park et al., applied CLASS (Closed-loop accumulation of single scattering) wavefront sensing to single molecule localization microscopy (SMLM) to correct aberrations. This results in convincing results in both artificially introduced aberration layers and tissue specimens such as brain sections and zebra fish.

The paper focusing on the application of CLASS technique to SMLM without much modification. In demonstrating CLASS for microtubule specimens where aberrations are introduced by artificial layer, gold nanoparticle is used to provide signal for CLASS. These aberrations are corrected effectively. For other tissue demonstrations, intrinsic reflection signals are used for CLASS. The average PSF shows the effectiveness of compensation.

I have a few comments on this manuscript:

1. Figure 2 demonstrated highest aberration amplitude corrected by CLASS. However, it requires the usage of gold nanoparticles because of the low reflectance of this specimen. If strong reflectance is required for CLASS to work, would this limit the application range of CLASS-SMLM to tissues containing large aberrations amplitudes? Would this technique work for thinner specimens, or would there be variations on CLASS performance for different specimens, depending on the amount of reflective signal which it could provide? If so, could the authors define when it would or would not work?
2. It would be helpful if the demonstration in Figure 2 does not require gold nanoparticles and thus provide a ‘true’ test for CLASS application in SMLM. Could one introduce refractive index mismatch between the artificial layer and cell layer to enhance the reflective signal or can the reflective signal be generated between the cell and cover slide interface?
3. CLASS requires characterizing 2800 SLM random patterns while collecting corresponding images for analysis. The total run time is about 5 mins (ref. 21). Is this still the case in this manuscript? These important parameters are not mentioned anywhere in the main text. It would be helpful to list them as these are key parameters for other researchers to determine the suitability of this approach towards their application. Also, would single feedback sufficient to correct the aberration. If not, how many iterations are used and for how long?
4. The manuscript needs to include characterizations of the CLASS-SMLM performance. What are the correction effectiveness at various levels of aberration in terms of wavefront error? What is its performance through index mismatched specimens at different depths?

5. The manuscript misses comparisons with the current state-of-the-art AO methods. Can CLASS-AO reach or surpass the level of wavefront error as previously demonstrated in SMLM? How much time would it take for a full AO compensation for CLASS-AO comparing with other SMLM-AO methods? Can the authors compare the wavefront errors from CLASS-AO with a previous SMLM-AO approach in some of their applications? The reviewer believes these additional comparisons would enhance the manuscript significantly.

As a direct application of CLASS-AO to SMLM, the manuscript contains convincing demonstrations for tissues and whole-mount animals. However, the manuscript is written in a way as an application note of CLASS. It lacks the explanations of the sensing concept and exact procedure in AO correction. How would these specific procedures affect the subsequent SMLM imaging. What type of specimen it would be mostly suited and which ones it would not. The current manuscript lacks performance characterizations (#4) and comparison with the state-of-the-art method in SMLM-AO (#5).

Manuscript #: NCOMMS-22-43570-T

Title: Label-free adaptive optics single-molecule localization microscopy for whole animals

Authors: Sanghyeon Park, Yonghyeon Jo, Minsu Kang, Jin Hee Hong, Sangyoon Ko, Suhyun Kim, Sangjun Park, Hae-Chul Park, Sang-Hee Shim, and Wonshik Choi

We are grateful to all the reviewers for their thorough and constructive opinions. In this response letter, we have addressed all the issues raised by the reviewers. In particular, we conducted extensive additional experiments and analyses directly comparing CLASS-SMLM with existing AO-SMLMs. Our results indicate that CLASS-SMLM is capable of correcting more than three times more complex aberrations and achieving greater imaging depth in zebrafish imaging. These findings are described in detail in ‘Note 1’ at the beginning of our response letter. Our responses to other major issues are summarized in the following.

- Emphasizing that our study is not a mere combination of CLASS and SMLM, but rather a significant advancement in overcoming the depth limitations of SMLM, which required additional efforts to combine these two delicate and complex imaging modalities.
- Providing detailed explanations on the working principles of CLASS-SMLM and why SMLM-based super-resolution imaging is more susceptible to aberrations than diffraction-limited imaging modalities.
- Presenting the time required for aberration measurement and correction using CLASS, as well as comparing the aberration correction speed per mode with existing AO-SMLM methods, indicating that CLASS-SMLM is almost 70 times faster.
- Conducting additional experiments to investigate the effect of the number of SLM control pixels on aberration correction accuracy.
- Conducting further experiments to explore the potential of the alternative tissue-clearing method suggested by the reviewer and the relevance of CLASS-SMLM in this context.

We have incorporated important results derived from additional experiments and analysis into the main text and Supplementary Information, with changes highlighted in red. We hope that the reviewers are satisfied with our responses. Once again, we appreciate their constructive comments, which have substantially improved the scientific integrity of our paper.

Note 1. Comparison of CLASS-SMLM with the current state-of-the-art AO-SMLM

In this note, we present experimental results comparing the aberration correction capabilities of our CLASS-SMLM method with those of existing AO-SMLM methods. Our results showed that the previous AO-SMLM could correct aberrations up to around 1 rad in RMS wavefront distortion, as reported in the literature. However, it was not able to correct the same aberration map that our CLASS-SMLM could correct when the wavefront distortion exceeds well beyond the 1-rad limit. These additional experiments confirmed the superior performance of the proposed approach.

[1] Aberration correction capability of the previous AO-SMLM

We chose REALM¹, the latest AO-SMLM modality in the field, as a point of reference. Following this earlier work, we prepared a monolayer of COS-7 cells and displayed a known aberration ϕ_{applied} on the spatial light modulator (SLM) in the emission beam path. We then used the REALM algorithm to find an aberration correction map. Essentially, the algorithm optimizes a designed image metric evaluated from blinking images of single molecules in the microtubules of COS-7 cells.

To check the fidelity of REALM, we measured the aberration correction map ϕ_{system} before applying ϕ_{applied} and ϕ_{total} after displaying ϕ_{applied} , respectively. Here, ϕ_{system} measures the aberration from the cells and SMLM setup. We compared the measured net aberration $\phi_{\text{net}} = \phi_{\text{total}} - \phi_{\text{system}}$ to ϕ_{applied} . If the measurement were perfect, ϕ_{net} would be equal to ϕ_{applied} .

To assess the degree of aberrations that REALM can handle, we gradually increased the RMS wavefront distortion of ϕ_{applied} from 1 rad to ~ 3 rad. We verified that REALM could find the aberration map faithfully when RMS wavefront distortion was 1 rad (Figure R1a). In this case, the aberration map was made of low-order Zernike modes (up to the 15th mode). ϕ_{applied} and ϕ_{net} were almost the same with a correlation of 0.970.

Under the same experimental condition of the cell monolayer sample, we applied the aberrations from brain tissues and zebrafish. We displayed an aberration map from the mouse brain tissue (0.983 rad in RMS wavefront distortion) and obtained a correlation of 0.895 between ϕ_{applied} and ϕ_{net} (Figure R1b). There was a slight decrease in correlation because real sample aberrations include high-order Zernike modes, which are not dealt with in the REALM algorithm. REALM showed worse performance for aberration maps with increased wavefront distortion. For zebrafish aberrations with RMS wavefront distortions of 2.14 rad and 3.08 rad, the correlations were 0.654 (Figure R1c) and 0.133 (Figure R1d), respectively.

Figure R1. Performance evaluation of REALM in finding the applied aberrations. Aberration maps ϕ_{applied} applied on the SLM and ϕ_{net} found by the REALM for four different levels of RMS wavefront distortions. **a** ϕ_{applied} is an artificial aberration made of Zernike modes used in REALM. Its RMS wavefront distortion is 1 rad. **b-d** Applied aberrations ϕ_{applied} are those in Fig. 3f (**b**; 0.983 rad), Fig. 4d (**c**; 2.14 rad), and Fig. 4g (**d**; 3.08 rad). Correlations are $C_a = 0.970$, $C_b = 0.895$, $C_c =$

0.654, and $C_d = 0.133$, where C_i represents the correlation between ϕ_{applied} and ϕ_{net} for $i = \mathbf{a}, \mathbf{b}, \mathbf{c},$ or \mathbf{d} .

As a next step, we conducted SMLM imaging of microtubules of COS-7 cells by correcting the applied aberration ϕ_{applied} with ϕ_{net} found by REALM (Figure R2). In other words, we displayed residual aberration $\phi_{\text{residual}} = \phi_{\text{applied}} - \phi_{\text{net}}$ on the SLM. We observed a gradual reduction in localization number and resolving power with the increase of the RMS wavefront distortion of ϕ_{applied} . In particular, almost all the microtubules were missing for 3.08-rad zebrafish aberration (Figure R2d). We would like to emphasize that our CLASS-SMLM worked well for this level of aberration even at a depth of 102 μm inside an intact zebrafish. On the contrary, REALM didn't work at all for the same aberration map for the cell monolayer samples where there were almost no depth-related signal loss and background fluorescence noise.

Note that we implemented REALM in a better setting than the original paper. Our aberration correction map was composed of 283,657 SLM pixels while the REALM paper used a deformable mirror (DM) with only 52 actuators¹. As we analyzed in detail, the number of correction elements plays an important role in the aberration correction (refer to Figure R7 for detailed discussion).

Figure R2. SMLM imaging by the REALM for COS-7 cells with the applied aberration ϕ_{applied} . **a-d** AO-off and -on SMLM images. AO-off SMLM images were obtained without AO. AO-on SMLM images were obtained by displaying residual uncorrected aberration $\phi_{\text{residual}} = \phi_{\text{applied}} - \phi_{\text{net}}$ on the SLM. (ϕ_{applied} and ϕ_{net} shown in Figure R1). RMS wavefront distortions of ϕ_{residual} were 0.246 rad (**a**), 0.570 rad (**b**), 1.22 rad (**c**), and 5.98 rad (**d**). Insets indicate ϕ_{residual} . $\sigma_{\text{AO off}}$ and $\sigma_{\text{AO on}}$ represent AO-off and -on localization precisions. $N_{\text{AO off}}$ and $N_{\text{AO on}}$ represent AO-off and -on localization numbers. Color bars indicate localization numbers. Scale bars indicate 5 μm .

[2] SMLM imaging of an intact zebrafish using the REALM

We verified the performance of REALM by imaging intact zebrafish embryos. We found that REALM worked to the extent that the measured aberration was as mild as 1 rad in RMS wavefront distortion (Figure R3a-c). While the aberration map identified by the REALM was not identical to the aberration map measured by CLASS, they have a reasonable correlation of 0.591. Referring to Figure R1c and Figure R2c (correlation was 0.654), this is high enough to explain the reasonable aberration correction by the REALM. However, when the aberration exceeded 2 rad in RMS wavefront distortion, the REALM didn't work at all (Figure R3d-f). There were significant degradation in localization precision and a reduction in localization number. The aberration correction map by the REALM was

completely different from that by the CLASS (Figure R3e, f). Their correlation was only 0.0688. Moreover, multiple trials of the REALM on the same region resulted in finding different aberrations (their correlations < 0.3), which confirms that the REALM completely failed.

As was presented in the manuscript, our CLASS-SMLM worked well in imaging an intact zebrafish even when the aberration exceeds 3 rad in RMS wavefront distortion. Therefore, our CLASS-SMLM is superior to the previous AO-SMLM in dealing with complex tissue aberrations. Note here that it is intrinsically not possible to conduct SMLM imaging for the same spot by both REALM and CLASS-SMLM. Most of the dyes were photobleached during raw image acquisitions with a REALM correction map, making it impossible to conduct another round of image acquisitions with a CLASS correction map. Therefore, we considered CLASS-identified aberration as a ground truth and compared it with the aberration map identified by REALM to see to what extent the REALM works properly.

Figure R3. SMLM images of oligodendrocytes in spinal cords of the 5-dpf intact zebrafish embryos by the REALM at a depth of 47 μm . **a** AO-off and -on SMLM images with REALM correction map in **b**. **b, c** Aberration correction maps found by REALM (**b**) and CLASS (**c**). **d-f** Same as **a-c**, respectively, for a different sample. $\sigma_{AO\ off}$ and $\sigma_{AO\ on}$ represent AO-off and -on localization precisions. $N_{AO\ off}$ and $N_{AO\ on}$ represent AO-off and -on localization numbers. Color bars indicate localization numbers. Scale bars indicate 5 μm . The RMS wavefront distortions of the aberration correction maps are 0.850 rad (**b**), 1.66 rad (**c**), 7.05 rad (**e**), and 2.18 rad (**f**). The correlation between **c** and **d** is 0.591, and that between **e** and **f** is 0.0688.

Reviewer #1

Review report on “Label-free adaptive optics single-molecule localization microscopy for whole animals” by Sanghyeon Park et. al., where the review is particularly focused on optical methods used in manuscript such as implementation of the wavefront sensing and details relating to the spatial light modulation presented in the manuscript.

This work reports the application of label-free wavefront sensing adaptive optics (AO) to single-molecule localization microscopy (SMLM) for deep-tissue super-resolution imaging. A number of investigations on highly aberrated specimens such as thick tissue slices and intact organisms with and without AO were presented and compared together. A closed-loop accumulation of single scattering (CLASS) microscopy was employed as a label-free wavefront-sensing AO method to determine the sample aberration based on intrinsic reflection signals. With the aid of AO-SMLM the authors were able to image thick (thickness of 150-200 μm) brain slices of Thy1-EGFP transgenic mice, where without AO system the SMLM imaging has suffered from shallow imaging depth of sub-100 μm thickness. The used methods for the imaging, wavefront sensing, and aberration removing are clearly presented in the manuscript or in the Supplementary Information. In my opinion results deserve publication. However, I have some comments on the manuscript and Supplementary Information. My comments are mostly focused on the implementation of the wavefront sensing and the optical methods used in the study, and hope to receive proper response before acceptance of the work.

We are grateful to the reviewer for recognizing the significant advancements in our study and endorsing its publication. We appreciate the thorough examination of our paper and the valuable feedback and suggestions provided. In this response letter, we clarified all the details of the implementation of the wavefront sensing and optical methods that the reviewer pointed out, which has greatly enhanced the overall quality of the paper.

SMLM is basically based on the fact that the spatial coordinates of single fluorescent molecules (also called emitters) can be determined with high precision if corresponding PSFs do not overlap. Subpixel shifts in the coordinates of an emitter lead to predictable changes in pixel intensities that can be used to compute its precise location. The localization precision reflects the scatter of localizations that would be obtained if a molecule was imaged and localized many times, and is basically limited by the signal to noise ratio (SNR) and not by the wavelength of light or the pixel size. To avoid overlaps between the PSFs of individual molecules, fluorescent emissions of distinct molecules are separated in time; the most common approach to obtain this temporal separation uses the phenomenon of photoswitching, where fluorescent molecules can switch between an active ‘ON’ (also called ‘bright’) state, where they emit fluorescent light when excited, and one or more inactive ‘OFF’ (also called ‘dark’) state in which they do not fluoresce. In the current work a label-free wavefront sensing adaptive optics method is applied on SMLM for deep-tissue super-resolution imaging. The details of processes used for improving PSFs of single molecules and differences with the conventional SMLM which is reviewed above should be emphasized.

We appreciate the reviewer's clear summary of the working mechanism of SMLM. In our study, we used conventional SMLM based on photoswitching, but we added a spatial light modulator (SLM) in its emission beam path to convert aberrated point spread functions (PSFs) caused by the tissue into near-diffraction-limited PSFs. Our study is unique in that we measure tissue-induced aberrations by recording reflected waves and applying the CLASS algorithm. Therefore, the aberration measurement is independent of fluorescence detection, unlike previous AO-SMLM techniques.

As the reviewer has summarized, the localization precision of the SMLM is determined by the PSF width σ_0 measured by the standard deviation of PSF, the number N of detected photons emitted from the molecules, and the signal-to-noise ratio (SNR). In the case of Gaussian PSF, the localization precision is given as

$$\sigma_{\text{loc}} = \sqrt{\left(\frac{\sigma_0^2 + a^2/12}{N}\right) \left(\frac{16}{9} + \frac{8\pi(\sigma_0^2 + a^2/12)b^2}{Na^2}\right)},$$

where a is the pixel width and b^2 is the expected number of photons per pixel due to background noises such as shot noise and fluorescence from other depths². To make the discussion concise, let us assume $a \ll \sigma_0$. Then, the localization precision is written as

$$\sigma_{\text{loc}} = \frac{\sigma_0}{\sqrt{N}} \sqrt{\frac{16}{9} + 8\pi \left(\frac{\sigma_0^2}{N}\right) \left(\frac{b^2}{a^2}\right)}.$$

Our proposed method reduces the PSF width σ_0 by a factor of 1.23-4.38 by correcting the aberration using the SLM in the emission beam path. This improves the baseline localization precision, σ_0/\sqrt{N} , by the same factor. The reduced PSF width enhances SNR by raising the photon number density, N/σ_0^2 . This attenuates the contribution of the background noise $8\pi \left(\frac{\sigma_0^2}{N}\right) b^2$, which is inversely proportional to SNR. In fact, the effect of the background noise was so detrimental in tissue and zebrafish imaging that localization has failed in many cases. Our CLASS-SMLM has enhanced the localization number by as many as 37.4 times by enhancing the SNR, thereby improving the resolving power.

The aberration correction also converts the non-Gaussian and often asymmetric PSFs into near-Gaussian PSF. Although it is difficult to quantify, correcting PSF shape helps to enhance the localization precision because the localization algorithms assume a Gaussian PSF and, thus, underperform for non-Gaussian PSFs. All these combined effects lead to enhanced localization precision and increased localization number.

We added the following sentences to the introduction to specify the aberration correction and SMLM imaging procedures.

“By physically correcting aberrations with an SLM in the emission beam path of SMLM, abnormal PSFs were restored to near-ideal PSFs. This led to the reduction in PSF width and enhancement of the SNR in SMLM imaging in which the centroids of individual single-molecule PSFs were fitted while they were randomly photoswitched.”

Actually, in the conventional SMLM, location of a single fluorescent molecule when adjacent molecules are ‘OFF’ can be precisely determined by determining the maximum intensity of the corresponding PSF even without using the PSF distribution. In the proposed AO-SMLM, AO helps to have a narrower PSF for a given single molecule when the other molecules’ PSFs do not overlapped. During whole area imaging of a sample by AO-SMLM, what arrangements were considered to prevent such overlapping of PSFs. Please clarify. Could you add some details about your SMLM. Do you use photoswitching? Do you use any fitting on the individual molecule PSF for more precise determining its location?

As the reviewer conjectured, we used conventional SMLM employing photoswitching to minimize the overlapping of PSFs. For inducing photoswitching, an imaging buffer was added to the sample chamber (see Methods for detailed preparation of the imaging buffer). The light-induced reactions of Alexa Fluor 647 with thiol cause the repeated switching of dye molecules between off and on states^{3,4}. In this process, some dye molecules are photobleached such that the spatial density of switched-on dyes continuously decreases. This leads to the reduction of the PSF overlap. Image acquisition gets started only after enough dye molecules are switched off and PSF overlap becomes negligible.

Gaussian fitting was used to find centroids of individual molecules’ PSFs. Specifically, we used ThunderSTORM, an open-source software in which each PSF is fitted with the following 2-D Gaussian function⁵.

$$h_G(x, y|\theta) = \frac{\theta_N}{2\pi[\sigma(\theta_z)]^2} \exp\left\{-\frac{(x - \theta_x)^2 + (y - \theta_y)^2}{2[\sigma(\theta_z)]^2}\right\} + \theta_0,$$

where $h_G(x, y|\theta)$: photon number at position (x, y) given the parameters $\theta = \{\theta_x, \theta_y, \theta_z, \theta_N, \theta_0\}$; $(\theta_x, \theta_y, \theta_z)$: molecular coordinates (what we want to find), θ_N : total number of photons emitted by the molecule, θ_0 : background offset.

We revised the following sentence in the Methods section to make it clear that the fitting of the single-molecule PSFs was used to form an SMLM image.

“All raw images of the blinking single-molecule emission PSFs were processed with the ThunderSTORM ImageJ plugin to find their centroids by the fitting.”

In the use of CLASS, still the resolution is limited by the diffraction limit. Is the super-resolution imaging advantage of the method only due to the use of AO-SMLM?

As explained earlier, CLASS-assisted aberration correction with an SLM reduces the PSF width σ_0 close to the diffraction limit. SMLM takes advantage of this PSF width reduction to achieve optimal super-resolution imaging.

It is necessary to emphasize that the restoration of abnormal PSFs to near-ideal PSFs of adjacent molecules are done at the different time? Or am I wrong?

The abnormal PSFs were restored to near-ideal PSFs for all the molecules within a certain area known as the isoplanatic patch. The isoplanatic patch is an area within which aberration stays almost the same. Its size depends on the type and internal structure of the sample. In the case of brain tissues, it was larger than the field of view (FOV) of the SMLM imaging ($\sim 33 \times 33 \mu\text{m}^2$). In the case of zebrafish, it was as small as $\sim 10 \times 10 \mu\text{m}^2$. Also, PSF correction is independent of time as the sample was fixed during the entire imaging session. We added these details to the “Label-free AO-SMLM setup” section in the main text.

In the current work, collected reflectance images from biological structures are used for evaluating the phase aberration induced by the tissue on the wavefront via CLASS microscopy, and the derived phase aberrations are successfully applied for imaging thick samples via AO-SMLM microscopy. Since phase aberration measurement from the reflectance images is one of the key parts of the current work, in my opinion it deserves and in some places it is needed to add details of the phase aberration deriving from the collected reflectance images. My main concerns are in this regard, which have been expressed below under comments on Supplementary Information, especially regarding Supplementary Figure SN7.

We thank the reviewer for pointing out the important aspect of our study in that the aberration is measured by intrinsic reflectance. In fact, we emphasized this point in the introduction in the following sentence.

“Using CLASS, we identified tissue aberration from the label-free measurement of the intrinsic reflectance signal of the tissue where single-molecule fluorescence was too weak for the detection or too aberrant for precise localization.”

It would be useful to determine and add some additional details about phase extraction method, such as the time needed for a given phase extraction, and the time needed to apply the necessary corrections in order to remove aberrations from the sample wavefront. How long does it take to capture an AO-SMLM image? Is it possible to apply the used method for imaging of a sample having dynamic feature?

CLASS calculation takes less than 7 seconds for an area of $\sim 33 \times 33 \mu\text{m}^2$, the FOV in the present study. Once aberration is evaluated, it takes less than a second to generate its correction map and display it on the SLM.

The acquisition time for SMLM imaging depends on the sample type. Usually, the cell monolayer required the shortest time of ~ 30 min (50,000 frames) because there was almost no scattering-induced photon loss. When imaging thick brain slices or intact zebrafish, a larger number of image frames were required due to the localization loss by the scattering-induced photon loss and background noise.

We added these details of the CLASS processing time and SMLM acquisition time in the Methods section.

CLASS imaging itself is fast enough to perform *in vivo* imaging as demonstrated in our earlier studies⁶. However, AO-SMLM is not ready for *in vivo* imaging yet due to the acquisition time for SMLM imaging in the tissue sample. Future developments in the designing of dye molecules and

improvements in detector sensitivity may allow *in vivo* SMLM imaging of living animals. In such cases, the aberration correction map should be updated continuously during the acquisition of raw images for SMLM imaging.

In most of the experimental figures there are some images of the samples having non-equal dimensions in the x and y directions, while the aberration correction maps were presented in circular frames. In which field of view the phase correction is applied? Is it cover all imaging area? Please add some clarification.

Aberration varies across the FOV ($\sim 33 \times 33 \mu\text{m}^2$), especially in the case of zebrafish imaging. From our analysis, the area where the aberration is relatively uniform is $\sim 10 \times 10 \mu\text{m}^2$. Therefore, we segmented the FOV and evaluated aberrations in each of $\sim 10 \times 10 \mu\text{m}^2$ subarea (Figure R4). For the acquisition of the CLASS-SMLM image, we chose one of the subareas where the structures of interest are located and applied the aberration correction for the corresponding subarea. We added this detail of the aberration correction procedure to the “Label-free AO-SMLM setup” section in the main text.

Figure R4. Correlations among aberration maps at individual subregions with respect to the central subregion. The FOV ($\sim 33 \times 33 \mu\text{m}^2$) in zebrafish data (Fig. 4g) was split into nine subareas. Aberration maps were obtained by applying CLASS algorithm for each subarea. Correlations among aberration maps at individual subregions are evaluated as $C_1 = 0.510$, $C_2 = 0.668$, $C_3 = 0.584$, $C_4 = 0.661$, $C_5 = 1$, $C_6 = 0.742$, $C_7 = 0.538$, $C_8 = 0.589$, and $C_9 = 0.574$, respectively, where C_i represents correlation between ϕ_i and ϕ_5 for $i = 1, 2, 3, \dots, \text{or } 9$.

In which range of angles the tissue structures were illuminated? What is the total number of images are used for a given aberration correction. Can you add details about applying different illumination angles?

To construct a reflection matrix for applying the CLASS algorithm, we scanned the angle of an incident planar wave in such a way as to fully cover the numerical aperture (1.2) of the objective lens. Therefore, the maximum illumination angle at the sample in water was approximately 71.6 degrees. Typically, we recorded 4,000 images to uniformly cover the pupil plane of the objective lens. The fidelity of finding the aberration depends on the number of illumination angles (Figure R5). We added the details of angular scanning to the Methods section.

Figure R5. Relation between image number and aberration. Aberration maps ϕ_N calculated from different number N of illumination angles ($N = 125, 250, 500, 1000, 2000$, or 4000). Correlations of each aberration with ϕ_{4000} are $C_{4000} = 1$, $C_{2000} = 0.955$, $C_{1000} = 0.885$, $C_{500} = 0.762$, $C_{250} = 0.0425$, and $C_{125} = 0.0340$, respectively, where C_N represents correlation between ϕ_N and ϕ_{4000} for $N = 125, 250, 500, 1000, 2000$, or 4000 .

In lines 161 and 162 it was mentioned that “Considering the microtubule diameter (approximately 25 nm) and size of primary and secondary antibodies (10-15 nm), this value agrees with well-known widths of microtubules.” Are all microtubules the same size?

The measured width of microtubules varies from point to point, as illustrated in Figure R6, and typically falls within the range of 60-80 nm.

Figure R6. Microtubule widths measured in an aberration-free COS-7 cell. FWHM of cellular microtubules were measured as 79 nm (a), 77 nm (b), 67 nm (c), 75 nm (d), 73 nm (e), 74 nm (f), 80

nm (g), 79 nm (h), 80 nm (i), 68 nm (j), 74 nm (k), 77 nm (l), 68 nm (m), 79 nm (n), 64 nm (o), and 77 nm (p). Color bar indicates localization numbers. Scale bar indicates 5 μm .

What is the number of wavefront cells of AO correction? How does it affect restored PSF?

We investigated how SLM pixel number affects restored PSF. For PSF evaluation, we imaged 100-nm-diameter fluorescent beads covered with an aberrating layer. Aberration was measured using reflections from pre-attached 100-nm-diameter gold particles. Without AO, the PSF was highly distorted due to the presence of substantial aberration (Figure R7a). Using CLASS-SMLM, we obtained the aberration correction map (bottom, Figure R7b), whose RMS wavefront distortion was estimated to be 1.78 rad. By displaying this correction map on the SLM, we could obtain a clean and bright PSF (top, Figure R7b). The number of SLM pixels used for this correction map was 283,657.

To verify the effect of the number of wavefront cells for aberration correction, we grouped several SLM pixels together and controlled them by designating their average value. We increased the number of pixels being grouped together on the x and y axes from 2^1 to 2^{10} . In other words, the spatial frequency resolution Δk of the aberration correction was varied from 2^1 to 2^{10} SLM pixels. Then, we recorded the aberration-corrected bead images while displaying individual correction maps (Figure R7c-e). As expected, the width and intensity of the PSF have deteriorated with the increase of Δk (Figure R7f-h). Notably, there was an abrupt increase of the PSF width from $\Delta k = 2^8$ SLM pixels (Figure R7g) while the PSF intensity decreased steadily (Figure R7h). This decrease is faster than normal bleaching (green curve in Figure R7h), making it clear that the intensity decrease resulted from uncorrected residual aberration. As a point of reference, we indicated the achievable level of aberration correction by the deformable mirror as black vertical lines in Figure R7g and h based on the number of actuators.

In conclusion, the reduction in the wavefront cells has a detrimental effect on the sharpness and brightness of the PSF. On the one hand, there is a specific number of wavefront cell number that can recover PSF width close to the diffraction limit for a given complexity of the aberration. On the other hand, the brightness of the PSF decreases steadily with the reduction of the wavefront cell number. Considering the importance of the PSF intensity in the localization of single molecules, especially for thick tissues, it is better to use as many wavefront cells as possible for optimal aberration correction.

Figure R7. Aberration correction with correction maps of different spatial frequency resolution.

a AO-off diffraction-limited image of a 100-nm-diameter bead with an aberration layer. Scale bar indicates 500 nm. **b** Same as **a**, but with AO (top), and the aberration correction map (bottom). **c-e** Same as **b**, but with the binning of the SLM pixels and thus varying the k-space resolution Δk to 2^4 (**c**), 2^6 (**d**), and 2^7 (**e**) SLM pixels, respectively. Scale bars indicate 500 nm. **f** Line profiles of the PSFs in **a-e** along the longest axis of AO-off image in **a**. **g** PSF FWHMs. FWHM of AO-off PSF is shown for reference. **h** Intensity change in bead images. The green line indicates decreasing intensity of 100-nm-diameter beads due to normal bleaching (without an aberration layer; without AO). Black vertical lines in **g** and **h** correspond to a deformable mirror with 25 actuators.

We added the following sentence to the Discussion section and incorporated the detailed results into the Supplementary Note.

“We found that the number of control elements used in the SLM plays an important role in determining the complexity of aberration that can be corrected and the width and brightness of the recovered PSFs (see Supplementary Note).”

A definition for “localization number” is needed.

The localization number is the number of single-molecule PSFs localized by the ThunderSTORM algorithm. It is determined by the number of center coordinates identified by the 2-D Gaussian fitting. We added this definition of the localization number to the revised manuscript.

It would be nice to introduce the signal to noise ratio (SNR) at the first use.

We added its definition to the first paragraph of the introduction.

In lines 176 and 177 change “A SMLM images of ...” to “a SMLM images of ...”.

We replaced capital “A” with a lower case “a” in Fig. 3.

Please add details about plots Figs. 2i-2l, Fig. 3j and Fig. 3j,k, or add some introduction to the nearest neighbor analysis.

We added “; see Methods” in the last paragraph of the section “Proof-of-concept imaging of microtubules in a cell through an aberrating layer” to the main text. Although Methods already contained a brief explanation of the nearest neighbor analysis, it was not referred to in the main text.

My comments on Supplementary Information:

In line 152, change “The 678-nm laser beam” to “The 678 nm laser beam”.

We removed the hyphen.

In supplementary Not, lines 160 – 163 you mentioned that “After being reflected from the reference mirror, RB retraced the same path back and traveled toward the grating through BS3. Passing through the grating, RB was diffracted into several branches. Among them, only the first-order branch was selected for oblique incidence of RB at the camera plane. The other branches were discarded by blocking with an iris.” and as is shown in Supplementary Figure SNI, you used a grating after BS3. In my opinion you can remove the grating and iris and only by a small rotation of the mirror used in the path, again you have the desired angle between RB and SB on the sCMOS camera. Please mention the reason for the use of grating?

Rotating a mirror tilts the temporal front of a wave as well as its propagation direction. For a light source with a short coherence length, if RB is tilted by a mirror, the interference takes place in a narrow intersecting area where the temporal fronts of SB and RB meet together (Figure R8b, d). Then, the resultant complex field map does not cover the entire FOV, missing sample information at most of positions. On the contrary, the first-order diffraction by the diffraction grating changes only the propagation direction, not the temporal front. Therefore, interference occurs throughout the entire FOV (Figure R8c, f) when RB is tilted by the grating.

Figure R8. Interference when the reference beam tilted by either a mirror or a grating. **b, c** Temporal wavefronts of SB (sample beam E_S ; colored in red) and RB (reference beam E_R ; colored in cyan) are shown. Here, the beam propagation direction was changed by either a mirror (**b**) or a grating (**c**). After being reflected from the sample, SB undergoes aberration-induced temporal wavefront distortion and then joins RB (See Fig. 1a or Supplementary Fig. SN1). RB has a flat temporal wavefront because it propagates through free space. **d, f** Complex field maps corresponding to **b** (**d**) and **c** (**f**), respectively. The saturation and the color bar indicate the amplitude and phase of the complex field, respectively. Scale bars indicate 20 μm . **e** Magnified view of white dashed box in **d**. [Quoted from Fig. 1 in Kim et al. Nature Communications **10**, 1 (2019)]

We added the following sentence to Supplementary Note 1.

“The first-order diffraction from the grating was used to set the temporal front of RB parallel to the camera plane. In case RB is tilted by rotating a mirror in the RB path, the tilt of the temporal front disrupts the formation of the interference pattern with SB.”

In the same figure, Supplementary Figure SN1, a mirror is missing after BS2 in the light return path.

We added missing mirrors in Supplementary Figure SN1.

In Supplementary Note, lines 164 – 165 you mentioned that “At the camera plane, SB and RB formed an interference pattern when the length difference between them was shorter than the coherence length.” There is no any additional information about SB reflected from the sample plane. Is it focused on a given point on the sample or it covers whole area of the sample? What is the size of area illuminated by the SB?

In the present study, the CLASS microscope has used planar waves of various incident angles as an illumination. Its beam size at the sample was set as $\sim 100 \mu\text{m}$ in radius. We specified this beam size in the Supplementary Note.

How does the aberrations of the sample affect and deviate the SB? Is SB still an almost plane wave with small phase fluctuations, after reflecting from the sample. Please clarify these. You have mentioned previously that the sample was illuminated widefield, but it is not clear that the phase aberrations of the sample how modulates entire SB wavefront?

A sample wave is the superposition of planar waves with different propagation angles. When it travels through an aberrating medium, the planar waves constituting the sample wave experience phase retardations depending on the propagation angle. This led to the distortion of the resulting sample wave. In the reflection configuration, the distortion of the sample wave occurs twice, one on the way in and the other on the way out. The spatial frequency of the sample wave can thus be described as

$$\tilde{E}(\mathbf{k}_{\text{out}}; \mathbf{k}_{\text{in}}) = \sqrt{\gamma} P_{\text{out}}^a(\mathbf{k}_{\text{out}}) \tilde{O}(\mathbf{k}_{\text{out}} - \mathbf{k}_{\text{in}}) P_{\text{in}}^a(\mathbf{k}_{\text{in}}) + \sqrt{\beta} \tilde{E}_M(\mathbf{k}_{\text{out}}; \mathbf{k}_{\text{in}}).$$

Here, $P_{\text{in}}^a(\mathbf{k}_{\text{in}}) = P(\mathbf{k}_{\text{in}}) \exp[i\phi_{\text{in}}(\mathbf{k}_{\text{in}})]$ and $P_{\text{out}}^a(\mathbf{k}_{\text{out}}) = P(\mathbf{k}_{\text{out}}) \exp[i\phi_{\text{out}}(\mathbf{k}_{\text{out}})]$ are respectively the input and output pupil functions in the presence of aberrations. $\phi_{\text{in}}(\mathbf{k}_{\text{in}})$ and $\phi_{\text{out}}(\mathbf{k}_{\text{out}})$ are the input and output aberrations, respectively, and $P(\mathbf{k})$ describes the ideal pupil function, which is unity when $|\mathbf{k}| \leq k_0\alpha$ and zero otherwise. Here, α is the numerical aperture of the objective lens. Our CLASS algorithm is unique in that it finds $\phi_{\text{in}}(\mathbf{k}_{\text{in}})$, $\phi_{\text{out}}(\mathbf{k}_{\text{out}})$, and $P(\mathbf{k})$ out of the reflection measurements. We added this detail to Supplementary Note.

What is the effect of depth in the sample on the aberrations, and how can consider the aberrations of different depths in the tissue.

In general, the deeper the imaging depth is, the stronger aberration is. However, it is not always the case. The degree of aberration depends highly on the local heterogeneities in the refractive index. For example, the aberration in Fig. 3a (depth: $\sim 50 \mu\text{m}$, RMS wavefront distortion: 1.37 rad) was stronger than that in Fig. 3f (depth: $\sim 74 \mu\text{m}$, RMS wavefront distortion: 0.983 rad). Also, the aberration in Fig. 4j (depth: $\sim 52 \mu\text{m}$, RMS wavefront distortion: 2.13 rad) was comparable to that in Fig. 4d (depth: $\sim 82 \mu\text{m}$, RMS wavefront distortion: 2.14 rad).

Aberration varies with the target imaging depth. We used time gating for measuring the aberration for a specific depth of interest. Time gating takes advantage of the fact that interference occurs when the optical path difference between two interfering beams is shorter than the coherence length Δ of the light source. This means that interference occurs within a depth range $z = z_0 - \Delta/2$ to $z_0 + \Delta/2$ where z_0 is the target imaging depth.

We added the following sentence to the Supplementary Note to specify how the depth of interest is set in the experiment.

“Due to its short coherence length ($\sim 40 \mu\text{m}$), it enabled time-gated detection in the interferometry. This provided depth gating defining the depth of interest.”

Did you image the reflected SB on the sCOMS camera?

Yes. We recorded the interference of the reflected SB with the RB by an sCMOS camera.

Do you have considered the specimen includes a transparency shield and a main structure, in which the sample beam is reflected from the main structure and double-passed through the transparent shield (which is mainly a non-homogenous and non-flat layer)?

The transparent shield that the reviewer is referring to is the tissue covering the main structure at a specific depth of interest. As the reviewer mentioned, the tissue is spatially inhomogeneous and non-flat such that it causes wavefront distortion. And the distortion occurs twice as explained above, one on the way in and the other on the way out.

Could you explain the origin of phase aberrations implemented on the reflected beam with more details?

When a planar wave propagates through a spatially heterogeneous tissue, it experiences phase retardation depending on the propagation angle. This angle-dependent aberration causes the distortion of the sample wave composed of multiple angular planar waves. The effect of aberration was clearly described in response to the reviewer’s earlier comment.

In lines 166-171, it is necessary to clarify the purpose for the point illumination of the sample and imaging of it on sCMOS through OL2?

The purpose of the point illumination is to record confocal reflectance images for roughly identifying the region of interest. Confocal reflectance imaging allows us to navigate the specimen and find the structure of interest in real-time. For this reason, the CLASS setup was designed for the dual-mode operation, one for the recording of the reflection matrix and the other for confocal reflectance imaging, as explained in Supplementary Note. We added the following sentence to the Supplementary Note to specify the reason the confocal reflectance imaging channel was added to the setup.

“Confocal reflectance imaging allows us to navigate the specimen and find the structure of interest in real-time.”

In line 170, is it ok still you call the reflected beam as RB?

We replaced “RB” with “SB” in the section “Details layout of experimental setup” in Supplementary Note.

In Supplementary Figure SN7a and especially according to its Fourier transform in Supplementary Figure SN7b, a linear fringes interference pattern (at least over the bright area) should be appear? Why such pattern does not appear? Can you show an enlarged pattern to present such interference fringes, even in an inset in Supplementary Figure SN7a?

There was a degradation of image resolution while converting a word file to a pdf. Diagonal interference fringe is well visible in the original word file.

Following the reviewer’s suggestion, we added a magnified view to Fig. SN7 to clearly visualize the interference pattern.

What was the sample? Is it a reflectance phase or amplitude test plate?

We imaged a USAF target [2”×2” Positive, USAF 1951 hi-resolution target (Edmund Optics, Stock number: #58-198)] in Supplementary Figure SN7.

Did you record the interference pattern with the aid of the reflected sample beam?

Yes. We acquired an interference image between SB and RB.

Could you determine the size of sample under study and size of area imaged by the system?

The sample dimensions (width×height×thickness) were different depending on sample types (mouse brain slices: ~1 cm×8 mm×(150-200 μm), intact zebrafish embryos: ~2.5 mm×500 μm×500 μm). The FOV of SMLM and CLASS was ~33×33 μm². We added the dimensions of the samples to the Methods section.

You scanned incident angle of the beam on the sample surface and record different interference patterns, what is the value of scanning angle step? How affect the value of scanning angle step on the resolution of phase aberration extraction?

The largest angle of the incident beam was 71.6° at the sample plane. The difference between adjacent angles was 1.79×10⁻²°. Simply speaking, the CLASS algorithm finds the aberration for each incident angle. Therefore, the finer the angle step is, the more complicated aberrations the CLASS algorithm can measure.

In Supplementary Figure SN7d, you mentioned that the color bar shows amplitude of the inverse Fourier transform of selected spectral component in c. Since you are interested to determine the phase aberration of the sample, it is a bit confusing. Please clarify.

Figure SN7d shows the amplitude map of the electric field, and there is a corresponding phase map. We showed only the amplitude map for brevity. We clarified this in the caption of Fig. SN7.

I think that you presented an infinite mode interference pattern of the RB and SB in Supplementary Figure SN7d, where there is no angle between the interfering beams (I can address to such interference pattern in [S. Rasouli, F. Sakha, and M. Yeganeh, “Infinite-mode double-grating interferometer for investigating thermal-lens-acting fluid dynamics,” Meas. Sci. Technol. 29,

085201 (1-10), 2018]). It seems that infinite mode interference patterns can be used in CLASS and omit one step from phase aberration process such as Supplementary Figure SN7b. In this case other phase extracting methods should be used.

Figure SN7d is not an interference image. Rather, it is the amplitude map of the complex field map obtained from the off-axis interference image. The suggested paper (Saifollah Rasouli et al. Meas. Sci. Technol. 29, 085201 (2018)) presented a double-grating interferometer for obtaining the complex map. In this respect, there seems no real gain in using the double-grating interferometer configuration because it is still necessary to apply the CLASS algorithm to separately find the input and output aberrations.

In lines 382 and 383 you mentioned that “At this moment, aberration map is obtained by reshaping phase array applied to the rows of $R(k_{out}, k_{in})$ into a square form (Supplementary Figure SN7j).” I am interested in to know how you establish a relation between different k_{in} or the corresponding illumination angle and different position on the sample area?

A planar wave with a specific transverse wavevector \mathbf{k}_{in} illuminates the entire sample area uniformly. Therefore, the wave has a uniform amplitude across the sample area while its phase has a ramp in the sample plane.

Reviewer #2

The paper presents a method for single molecule localization microscopy that attempts to correct the aberrations induced by tissue scattering. Results are shown on both brain tissue slices and whole zebrafish. The results indicate a modest improvement in resolution and ability to image deeper in the presence of tissue scattering.

We appreciate the reviewer's critical and constructive comments. We have taken them into careful consideration and addressed them all in this revision. In particular, we have conducted extensive experiments and analyses directly comparing our CLASS-SMLM with existing AO-SMLM methods and demonstrated that our CLASS-SMLM outperforms existing AO-SMLM methods in correcting complex aberrations. Moreover, we have emphasized the importance of correcting aberrations in SMLM imaging, given its high sensitivity to aberrations compared to diffraction-limited imaging modalities.

Novelty: The main concern/limitation of the work as presented is that it is not clear what is new/novel about their technique. Essentially the paper completely relies on CLASS (a very elegant and interesting past work) to estimate aberrations in moderately scattering specimens -- and then just uses phase conjugation to correct for these measured aberrations. The only reason for CLASS as a technique to be interesting is to correct for aberrations -- and even in their earlier papers they already demonstrate that. The only new part seems to be that in this paper they use it for single molecule localization microscopy --- but that's straightforward in the cases where CLASS works.

We would like to emphasize that our study makes an important breakthrough in SMLM imaging in terms of imaging depth rather than a simple application of CLASS to SMLM imaging. SMLM is a powerful super-resolution imaging technique that enables the visualization of minute biological structures with superior spatial resolution than the diffraction limit. SMLM has proven to be a valuable tool for important biological studies, including but not limited to imaging synaptic structures, visualizing molecular complexes, discovering new structures, and investigating specific genes such as Alzheimer's disease⁷⁻⁹.

Despite its advantages, SMLM is highly sensitive to sample-induced aberrations and background noise, as it relies on fitting the weak emission PSFs from individual molecules. Any perturbations of the PSFs can greatly degrade the SNR and compromise the fitting process. As a result, the imaging depth of SMLM has been limited to a shallow range. Many notable efforts have been made to overcome this limitation^{1,10-12}, but these previous studies have an intrinsic limitation since they correct single-molecule PSFs based on their weak fluorescence emission signal.

In our study, we present a formidable solution to this challenge in SMLM by introducing CLASS microscopy, a method for measuring aberrations based on interference images from intrinsic reflection from samples. Unlike other AO-SMLM methods, CLASS-SMLM enables the measurement of aberrations even when the single-molecule PSFs are highly distorted, making it possible to correct wavefront distortion more than three times stronger than the previous 1-rad limit in existing AO-SMLMs. In fact, we had to design a setup in such a way as to precisely merge the CLASS microscopy and SMLM. We tuned the position of the tube lens in the commercial microscope to set the image relay from the sample to the camera to a 4-f configuration, which is essential for CLASS microscopy. Also, the SLM in the emission beam path of the SMLM was precisely calibrated to match the pupil plane of the CLASS microscopy.

All these efforts enabled us to achieve the deepest SMLM imaging depth of ~102 μm in an intact zebrafish. Our method led to a significant improvement in the localization number, up to 37-fold enhancement, compared to the typical 2-8 times enhancement seen in previous studies^{1,10}, while maintaining localization precision comparable to that of aberration-free samples. In doing so, we achieved spatial resolutions of 34 nm in a cell monolayer, 53 nm in brain tissues, and 63 nm in zebrafish (see Supplementary Figure 4), which are far beyond the diffraction limit. We believe that these advances highlight the importance of our work and justify its publication in a prestigious journal like *Nature Communications*.

Experiments: Experimental results indicate a slight improvement quantitatively --- about a 2-3x improvement in resolution compared to non-AO which indicates that the level of scattering in the samples are quite small to begin with.

We respectfully disagree with the reviewer's assertion that our results show only a slight improvement. Our study focuses on correcting the sample-induced aberration, which has been the primary factor limiting SMLM imaging. Previous AO-SMLM methods were limited to correcting RMS wavefront distortions up to 1 rad, beyond which even small perturbations cause significant impairment to the achievable resolution. In our study, we were able to correct aberrations as large as 3.08 rad in RMS wavefront distortion. This led to significant improvements in localization number enhancement, up to 37.4-fold, corresponding to Nyquist resolution enhancement of ~ 6.12 fold, and localization precision improvement of up to 3.61-fold.

While the reviewer may be interested in addressing scattering, it is important to note that our study is focused on correcting the sample-induced aberration, which is the major limiting factor for deep-tissue SMLM imaging. However, we agree that for further improvements in imaging depth, the correction of scattering could be beneficial.

We added the following sentence to the introduction to emphasize the difficulty in realizing SMLM imaging inside the aberrating tissues.

“Even weak aberration for diffraction-limited imaging modalities can have detrimental effects on SMLM imaging.”

The authors suggest that previous SMLM techniques operate with wavefront aberrations of the order of 1 radian and suggest that their proposed technique can handle much larger aberrations but their bio-samples are not all that much more scattering. The brain sample has a RMS wavefront distortion of ~ 1 , while the zebrafish it is ~ 2 .

Once again, we want to emphasize that SMLM is much more susceptible to aberrations than diffraction-limited imaging. This is because SMLM relies on weak fluorescence signals from individual molecules for image formation. Even a mild 1-rad aberration makes it difficult to resolve sub-diffraction-limited structures such as dendritic spine necks in SMLM images, as shown in Fig. 2 (Figure R9). Consequently, 2-3 times stronger aberrations in zebrafish had an even more detrimental effect on SMLM imaging. In these cases, single-molecule PSFs were not only blurred but also distorted in shape, making it much more difficult to fit the centroids. As a result, some of the structures were completely invisible due to the failure of localization (Figs. 4j, m). Our results showed that the correction of the sample-induced aberrations restores the PSFs and thus enables super-resolution imaging of whole organisms with localization precisions close to those of aberration-free samples.

Figure R9. Ensemble-averaged PSFs. a, c AO-off and -on ensemble-averaged PSFs for Figs. 3f (a)

and 4d (c), respectively. Scale bars indicate 500 nm. **b, d** x- and y-directional line profiles of PSFs for **a (b)** and **c (d)**, respectively. x' and y'-direction are defined as the longest axis and its orthogonal axis of the ensemble-averaged PSF. FWHMs of PSFs are written above line profiles.

Furthermore, the quantitative comparisons seem to be limited to the insets rather than averaged over the entire sample. The insets in all experiments seem to have been chosen to highlight the difference between nonAO and their method (which is understandable and what one would expect of authors). But one would expect that quantitative metrics of comparison such as resolution enhancement etc are averaged over the entire sample rather than be calculated and presented only on these author-selected regions. This makes it difficult to evaluate what the actual improvements in performance are.

We appreciate the reviewer's remark. In fact, we presented important parameters including localization precisions over the entire view field as well as specific ROIs. All of them are summarized in Supplementary Table 1.

Claims: In many places, the paper overclaims its contributions vis-a-vis the state of the art. The "whole animals" in the title should be replaced with zebrafish or otherwise toned down.

Following the reviewer's suggestion, we replaced "animal(s)" or "organism(s)" with "zebrafish" in all places including the title.

After all one particular reason the zebrafish is such a well studied model organism for optical microscopy techniques is because it has so little scattering compared to most other animal tissue.

As explained above, SMLM is an extremely sensitive technique. Even a relatively weak aberration can make a significant impact on its performance, unlike the diffraction-limited imaging modalities. Consequently, conducting SMLM imaging in small animals such as zebrafish has been a challenge. The non-flat body morphology and heterogeneous internal structures of these organisms can induce wavefront distortions of ~2-3 radians in RMS, greatly undermining the ability to localize single molecules for SMLM image formation. However, our proposed method could effectively correct this level of aberrations, enabling imaging inside the zebrafish at an unprecedented depth of ~102 μm .

Lack of comparisons: As the paper readily points out in their introduction, there have been several techniques that have been developed for AO in single molecule localization microscopy. But the current manuscript does not directly compare with any of them at all. Instead the quantitative comparisons are only with baseline SMLM without any adaptive optics, which is really not a fair baseline. In order for the paper to demonstrate an improvement over current state of art, direct comparisons with current state of art AO methods in SMLM need to be shown. Just a qualitative claim that current methods have not previously handled aberrations greater than 1 rad is not sufficient at all.

The reviewer has raised an important point regarding the need to directly compare our CLASS-SMLM with current state-of-the-art AO methods in SMLM. Following the reviewer's suggestion, we have conducted additional experiments and analysis to facilitate a direct comparison. To achieve this, we have implemented a recent method in the field, called REALM¹, in our system and verified its aberration correction capability.

Our results indicate that REALM can correct aberrations up to approximately 1 rad in RMS wavefront distortion, as reported in the literature. However, it was not able to correct the same aberration map that our CLASS-SMLM could correct when the wavefront distortion exceeds far beyond 1 rad. These additional experiments have confirmed the superior performance of our newly proposed approach. For a detailed comparison, please refer to Note 1 at the beginning of this response letter.

We added the content in Note 1 to the Supplementary Note and inserted the following phrase into the Discussion section for guiding readers to this new addition.

"(see Supplementary Note for detailed comparison between our method and previous AO-SMLMs)"

Reviewer #3

In the present manuscript, Park et al applied label-free wavefront sensing adaptive optics to SMLM (single-molecule localization microscopy) for deep-tissue super-resolution imaging. The authors succeeded in resolving sub-diffraction morphologies of cilia and oligodendrocytes in whole mount zebrafish as well as dendritic spine in thick mouse brain tissues at a considerable depth (around 100 micro-meter). The authors claim that the approach can expand the application range of SMLM to intact animals that cause the loss of localization points owing to severe tissue aberrations.

We deeply appreciate the reviewer's valuable opinions. In this revision, we have addressed all the issues raised through additional experiments and analysis. The suggestion to combine tissue clearing with high-resolution fluorescence microscopy is intriguing. However, as demonstrated in our additional experiments, there is still a significant residual aberration even with tissue clearing that can adversely affect SMLM imaging. Moreover, high-resolution fluorescence microscopy such as AiryScan does not enhance the resolution comparable to the achievable resolution with SMLM. Therefore, we believe that our study's contribution to extending the imaging depth of SMLM deserves recognition.

Major comments:

1) While it is a good idea to apply label-free CLASS to AO, an in vivo imaging with CLASS and an AO-scanning microscopy has already been reported by the same authors (Nature Communications, 2019). In this sense, I don't think there is a satisfactory conceptual leap. With this reason, I am not convinced that the manuscript is enough for Nature Communications. I feel that the manuscript would be suitable for publication in a more specialized journal.

We would like to emphasize that our study makes an important breakthrough in SMLM imaging in terms of imaging depth rather than a simple application of CLASS to SMLM imaging. SMLM is a powerful super-resolution imaging technique that enables the visualization of minute biological structures with superior spatial resolution than the diffraction limit. SMLM has proven to be a valuable tool for important biological studies, including but not limited to imaging synaptic structures, visualizing molecular complexes, discovering new structures, and investigating specific genes such as Alzheimer's disease⁷⁻⁹.

Despite its advantages, SMLM is highly sensitive to sample-induced aberrations and background noise, as it relies on fitting the weak emission PSFs from individual molecules. Any perturbations of the PSFs can greatly degrade the SNR and compromise the fitting process. As a result, the imaging depth of SMLM has been limited to a shallow range. Many notable efforts have been made to overcome this limitation^{1,10-12}, but these previous studies have an intrinsic limitation since they correct single-molecule PSFs based on their weak fluorescence emission signal.

In our study, we present a formidable solution to this challenge in SMLM by introducing CLASS microscopy, a method for measuring aberrations based on interference images from intrinsic reflection from samples. Unlike other AO-SMLM methods, CLASS-SMLM enables the measurement of aberrations even when the single-molecule PSFs are highly distorted, making it possible to correct wavefront distortion more than three times stronger than the previous 1-rad limit in existing AO-SMLMs. In fact, we had to design a setup in such a way as to precisely merge the CLASS microscopy and SMLM. We tuned the position of the tube lens in the commercial microscope to set image relay from sample to the camera to a 4-f configuration, which is essential for CLASS microscopy. The SLM in the emission beam path of the SMLM was precisely calibrated to match the pupil plane of the CLASS microscopy.

All these efforts enabled us to achieve the deepest SMLM imaging depth of ~102 μm in an intact zebrafish. Our method led to a significant improvement in the localization number, up to 37-fold enhancement, compared to the typical 2-8 times enhancement seen in previous studies^{1,10}, while maintaining localization precision comparable to that of aberration-free samples. In doing so, we achieved spatial resolutions of 34 nm in a cell monolayer, 53 nm in brain tissues, and 63 nm in zebrafish (see Supplementary Figure 4), which are far beyond the diffraction limit. We believe that these advances highlight the importance of our work and justify its publication in a prestigious journal like *Nature Communications*.

2) The aberration correction maps shown in Figures 1, 3, and 4 are inferior to those presented in the previous studies (Figure 4, Nature Communications 2017; Figure 2, Nature Communications 2019). This is presumably because the corrections were made in a large field. If this is the case, the authors need to state this clearly.

The primary reason why the aberrations in our study were weaker than those in earlier CLASS studies, which are diffraction-limited imaging modalities, is that the imaging depth of our CLASS-SMLM was shallower. In previous CLASS studies, reflectance images were acquired at a depth of 160 μm in the zebrafish hindbrain⁶. In contrast, the achieved depth of SMLM imaging in our study was ~ 100 μm .

It is worth noting that image formation by single-molecule localization is much more susceptible to background noise and aberrations than diffraction-limited imaging modalities. Consequently, even if the aberration was milder than in earlier studies, its effect was so detrimental that SMLM imaging did not work well without aberration correction. To emphasize this point clearly, we added the following sentence to the Introduction section.

“Even weak aberration for diffraction-limited imaging modalities can have detrimental effects on SMLM imaging.”

We would like to emphasize once again that our CLASS-SMLM overcomes the limitations of existing AO-SMLMs in terms of the complexity of aberration (three times stronger than the previous 1-rad limit) and the achievable imaging depth (twice as deep as the previous 50- μm limit)¹.

Although we were able to achieve an imaging depth of up to ~ 102 μm , this is not the ultimate depth limit of CLASS-SMLM. Maintaining focus is crucial in SMLM imaging because image acquisition typically takes a long time (≥ 30 minutes). We discovered that there is a depth limit to the focus maintenance system in our microscope, which is the Perfect Focus System (PFS) of the Nikon Eclipse Ti2-E microscope. In our current setup, its maximum working depth was 100-120 μm . If this technical issue is resolved in the future, there is room to further increase the imaging depth of CLASS-SMLM.

3) The authors applied the technique to fixed samples, not live animals. For fixed samples, tissue-clearing (with Scale, Clarity, SeeDB, Cubic, etc) is much more convenient and powerful than AO, and thus, the authors first need to employ tissue-clearing. After doing so, the authors need to evaluate the power of the method described in the present manuscript. Otherwise, the results cannot be evaluated properly in a practical sense. Also, for the evaluation, the authors need to compare the images of AO with images obtained by a commercially available high-resolution microscope such as Zeiss AiryScan.

We appreciate the reviewer’s interesting remark. In response to this suggestion, we employed tissue clearing for zebrafish embryos and conducted a closer examination of the specimen. For tissue clearing, we used the Binaree Tissue ClearingTM Kit (Cat #. HRTC-001) and Binaree Mounting and StorageTM Solution (Cat #. SHMS-060), following the procedures described in the literatures^{13,14}.

As illustrated in Figure R10, we discovered that there were still comparable levels of aberrations even after tissue clearing. The aberrations measured at depths of 72 μm and 102 μm were 1.17-1.90 rad and 1.38-2.91 rad, respectively, in RMS wavefront distortion. This explicitly suggests that aberration correction will remain a critical issue for SMLM imaging even after tissue clearing. Although tissue clearing can reduce photon loss by decreasing tissue scattering, it does not significantly reduce aberrations, most likely because wavefront distortion primarily originates from complex surface morphology.

Figure R10. Aberrations of a tissue-cleared intact zebrafish. **a, b** Confocal reflectance images of a tissue-cleared intact zebrafish at different depths. Due to clearing, the contrast of the reflectance image was significantly reduced. Scale bars indicate 50 μm . **c, d** Aberration maps measured at yellow and green boxes in **a** and **b**, respectively. Each aberration map corresponds to a subregion of $\sim 10 \times 10 \mu\text{m}^2$.

We would like to address some potential concerns regarding the use of tissue clearing in SMLM imaging. Tissue clearing has been shown to have several side effects. For instance, previous studies have reported that tissue clearing can result in changes in tissue size and morphology^{15,16}. This can be especially problematic for SMLM imaging, which requires high-quality images with minimal distortion.

Furthermore, the use of tissue-clearing solutions during imaging can also affect the refractive index of the sample, potentially disrupting index-matching processes that are essential for astigmatism-based 3D SMLM imaging¹⁷. In addition, tissue clearing can cause physical damage to the specimen, including broken blood vessels and non-uniform size changes¹⁸.

Finally, tissue clearing is a time-consuming process, with protocols such as CUBIC taking up to several days^{16,19}, and even longer protocols like PACT taking up to 24 days¹⁵. Given these challenges, it is not surprising that previous AO-SMLM studies have not used tissue clearing.

Therefore, while tissue clearing can reduce tissue scattering and photon loss in SMLM imaging, its potential drawbacks should be carefully considered before using it for SMLM imaging, particularly in cases where high-quality imaging with minimal distortion is essential.

We appreciate the suggestion to compare CLASS-SMLM with high-resolution imaging modalities like the Zeiss AiryScan. Unfortunately, however, we were unable to directly compare images from both systems as we could not locate the AiryScan microscope in our vicinity. Instead, we can provide a discussion comparing the two systems. The Zeiss AiryScan microscope can enhance spatial resolution up to 1.7 times compared to a confocal microscope using a pinhole size of 1.0 AU²⁰. This corresponds to a lateral resolution of 120-160 nm according to the Zeiss website. The achievable resolution will be poorer than this when the tissue aberrations exist. In contrast, the spatial resolution achieved in our CLASS-SMLM experiments is superior to that achievable by the AiryScan microscope. Even with the presence of aberrations, we achieved spatial resolutions of 34 nm in a cell monolayer, 53 nm in brain tissues, and 63 nm in zebrafish (see Supplementary Figure 4). These resolutions are 1.90-4.71 times higher than that achievable by the AiryScan microscope.

We added the tissue clearing results to the Supplementary Note and also inserted the following sentence in the Discussion section.

“One may consider employing tissue clearing to reduce scattering, but there still exists the need for correcting aberrations (see Supplementary Note).”

4) Fixed samples cannot be called “intact animals”. If the authors want to use the term “intact animals”, samples need to be alive. Upon applying the technique to live intact animals, it would be critical how much time is needed for the correction.

The definition of "intact" can vary depending on the field of research. In our study, we used the term to indicate that the samples were not sliced with a vibratome or cryotome, thus preserving their structural integrity. This terminology has also been used in earlier SMLM studies, such as Jeongmin Kim et al.'s work²¹ on oblique-plane single-molecule localization microscopy for tissues and small intact animals. They referred to fixed *C. elegans* and a fixed stickle fish as "small intact animals."

We partially agree with the reviewer's suggestion that it would be interesting to extend the capabilities of SMLM imaging to live animals. However, this is currently not feasible with our implementation. It takes around 35 seconds to acquire raw images for CLASS imaging, and additional time is required for aberration calculation (less than 7 seconds) and correction map uploading (less than 1 second). Furthermore, the actual SMLM imaging process takes more than 30 minutes. Addressing these limitations would be an exciting future direction to enable SMLM imaging of live animals.

We added the times taken for the aberration correction to the Methods section.

“The recording of complex field maps takes 35 seconds, and CLASS calculation takes less than 7 seconds for an area of $\sim 33 \times 33 \mu\text{m}^2$, the FOV in the present study. Once aberration is evaluated, it takes less than a second to generate its correction map and display it on the SLM.”

Reviewer #4

Park et al., applied CLASS (Closed-loop accumulation of single scattering) wavefront sensing to single molecule localization microscopy (SMLM) to correct aberrations. This results in convincing results in both artificially introduced aberration layers and tissue specimens such as brain sections and zebra fish.

The paper focusing on the application of CLASS technique to SMLM without much modification. In demonstrating CLASS for microtubule specimens where aberrations are introduced by artificial layer, gold nanoparticle is used to provide signal for CLASS. These aberrations are corrected effectively. For other tissue demonstrations, intrinsic reflection signals are used for CLASS. The average PSF shows the effectiveness of compensation.

We appreciate the thorough review comments provided by the reviewer. In this revision, we have provided detailed and comprehensive responses to each issue raised. Specifically, we have included an explanation of how the performance of CLASS varies depending on its complexity and target depth. Furthermore, we conducted direct comparison experiments between CLASS-SMLM and existing AO-SMLMs, which demonstrated that CLASS-SMLM is superior in compensating for complex aberrations. We have also analyzed the required time and correction speed of the CLASS-SMLM, compared to those of existing AO-SMLM. All of these revisions have significantly enhanced the scientific integrity of our paper.

I have a few comments on this manuscript:

1. Figure 2 demonstrated highest aberration amplitude corrected by CLASS. However, it requires the usage of gold nanoparticles because of the low reflectance of this specimen. If strong reflectance is required for CLASS to work, would this limit the application range of CLASS-SMLM to tissues containing large aberrations amplitudes? Would this technique work for thinner specimens, or would there be variations on CLASS performance for different specimens, depending on the amount of reflective signal which it could provide? If so, could the authors define when it would or would not work?

As pointed out by the reviewer, intrinsic reflectance is required for CLASS microscopy to identify aberrations. However, even through intact mouse skulls, reflections from brain tissues are strong enough to identify complex aberrations²². The working conditions for CLASS have been well documented in our previous studies²³, and we provide the essential part here.

The fidelity of aberration correction is determined by the parameter χ defined as $\chi = \xi(I_S/I_M)\sqrt{N}$. Here, ξ denotes the normalized cross-correlation of the aberration map quantifying the complexity of aberration, and its magnitude is decreased with the increase of the complexity of the aberration. N represents the number of orthogonal free modes detected at the camera. It is determined by the number of diffraction-limited spots within the recorded FOV, and it amounts to 2,285 in our experiment. I_S and I_M correspond to the average intensity of the signal wave and multiple scattering noise at each detection mode, respectively. CLASS works when χ is larger than a certain threshold χ_{th} , which varies depending on the system parameters and ingenuity of the algorithm.

In fact, CLASS is not required in cell monolayer as the aberrations are either negligible or correctable by existing methods. The reason we conducted cell monolayer imaging was to verify the performance of CLASS-SMLM in well-controlled conditions.

We added the following sentence to the Results section to emphasize that CLASS microscopy works well with intrinsic tissue reflectance. We also added the fidelity of the CLASS algorithm described above to Supplementary Note.

“In fact, CLASS microscopy is so sensitive that even weak intrinsic reflection signals from brain tissues under the mouse skull are strong enough to measure the aberrations.”

2. It would be helpful if the demonstration in Figure 2 does not require gold nanoparticles and thus provide a ‘true’ test for CLASS application in SMLM. Could one introduce refractive index mismatch between the artificial layer and cell layer to enhance the reflective signal or can the reflective signal be generated between the cell and cover slide interface?

As discussed in our response to comment #1, we conducted cell monolayer experiments to verify the performance of CLASS-SMLM under well-controlled, ideal conditions. Specifically, we could conduct SMLM imaging for microtubules, which are well-known structures^{24,25}, with negligible scattering-induced photon loss. The complexity of aberration could be controlled systematically in preparation for the artificial aberration layer. We believe that further characterization studies would not be necessary for cell monolayers since the performance of CLASS-SMLM was verified well enough in the brain slices and intact zebrafish embryos.

3. CLASS requires characterizing 2800 SLM random patterns while collecting corresponding images for analysis. The total run time is about 5 mins (ref. 21). Is this still the case in this manuscript? These important parameters are not mentioned anywhere in the main text. It would be helpful to list them as these are key parameters for other researchers to determine the suitability of this approach towards their application. Also, would single feedback sufficient to correct the aberration. If not, how many iterations are used and for how long?

In our system, it takes approximately 35 seconds to acquire interference images for 4,000 different angles with a FOV of $\sim 33 \times 33 \mu\text{m}^2$. This is much faster than the ~ 5 minutes reported by Kang, S. et al. (Nat. Commun. 8, 2157 (2017)) because our CLASS-SMLM system uses a faster galvanometer mirror for angle scanning instead of an SLM. We did not make an effort to further reduce this time because it is negligible compared to the acquisition time of single-molecule fluorescence images for SMLM, which can take over 30 minutes, especially for deep-tissue imaging. However, there is still room for reducing image acquisition time for CLASS. For example, we used a high-speed camera (Cheetah 800, Xenics; 6.8 kHz frame rate) for much faster image acquisition and successfully verified in vivo imaging of a living mouse²⁶. The CLASS algorithm runs only once and makes a few rounds of computational iterations to find the aberration from the reflection matrix. Therefore, there is no need for hardware iteration in CLASS-SMLM experiments, unlike the previous AO-SMLM methods.

We added the following sentences to the Methods section to provide key parameters that the reviewer has mentioned.

“Typically, we recorded 4,000 images to uniformly cover the pupil plane of the objective lens.”

“The recording of complex field maps takes 35 seconds, and CLASS calculation takes less than 7 seconds for an area of $\sim 33 \times 33 \mu\text{m}^2$, the FOV in the present study. Once aberration is evaluated, it takes less than a second to generate its correction map and display it on the SLM.”

“The acquisition time for SMLM imaging depends on the sample type. Usually, the cell monolayer required the shortest time of ~ 30 min (50,000 frames) because there was almost no scattering-induced photon loss. When imaging thick brain slices or intact zebrafish, a larger number of image frames were required due to the localization loss by the scattering-induced photon loss and background noise.”

4. The manuscript needs to include characterizations of the CLASS-SMLM performance. What are the correction effectiveness at various levels of aberration in terms of wavefront error? What is its performance through index mismatched specimens at different depths?

The ability of the CLASS algorithm to correct aberrations depends on the complexity of aberration, with higher degrees of aberration resulting in lower correction fidelity. The parameter χ , as explained earlier, determines the performance of the algorithm, and the parameter ξ therein is determined by the level of aberration. In addition, the number of SLM pixels displaying the correction map also affects the correction fidelity. If the number of pixels is smaller than the number of orthogonal modes, which is equal to the number of pixels (2,285 pixels) in a CLASS-evaluated aberration map, the correction fidelity will be reduced. To increase the fidelity of wavefront correction, we interpolated the original aberration map to 283,657 pixels and used as many SLM pixels.

We also conducted additional experiments to investigate the effect of the number of wavefront control pixels to the aberration correction capability (see Figure R7 and Supplementary Note). We found that the reduction in the control pixel number has a detrimental effect on the sharpness and brightness of the PSF. On the one hand, there is a specific control pixel number that can recover PSF width close to the diffraction limit for a given complexity of the aberration. On the other hand, the brightness of the PSF decreases steadily with the reduction of the control pixel number.

All of the samples we investigated had spherical aberration components due to the index mismatch between the immersion medium (water) and tissues. These components were already corrected in our CLASS-SMLM imaging. To illustrate this, we present the spherical aberration component in each of the experiments (Figure R11). We added this new analysis to the Supplementary Note.

Figure R11. Spherical aberration component in each sample aberration. a-g Sample aberrations (left) and their spherical aberration components (right). Their RMS wavefront distortions are written above aberration maps.

5. The manuscript misses comparisons with the current state-of-the-art AO methods. Can CLASS-AO reach or surpass the level of wavefront error as previously demonstrated in SMLM? How much time would it take for a full AO compensation for CLASS-AO comparing with other SMLM-AO methods? Can the authors compare the wavefront errors from CLASS-AO with a previous SMLM-AO approach in some of their applications? The reviewer believes these additional comparisons would enhance the manuscript significantly.

We appreciate the reviewer's valuable suggestion. We have implemented the reviewer's suggestion by conducting additional experiments and analysis to make a direct comparison between CLASS-SMLM and previous AO-SMLM methods (see Note 1 at the beginning of this response letter for details). Specifically, we have implemented the latest and most advanced AO-SMLM method, called REALM¹, in our system. Our results demonstrate that CLASS-SMLM outperforms REALM in both the extent of aberration correction and imaging depth. REALM is limited by a maximum RMS wavefront distortion of 1 rad or a depth of ~50 µm. Similarly, D. Burke et al. reported in *Optica* 2, 2 (2015) that they corrected aberration up to 0.88 rad. In contrast, our CLASS-SMLM was able to correct 3.08-rad aberration at a depth of ~102 µm in intact zebrafish.

In CLASS-SMLM, a complete AO compensation involves three steps. The first step is acquiring interference images, which takes approximately 35 seconds (~21 seconds for GV scanning + 14 seconds for image saving) to obtain 4,000 images for a ~33×33 µm² FOV. The second step is aberration calculation, which takes less than 7 seconds, and the third step involves creating and displaying a correction map, which takes less than 1 second. Therefore, the total time for 2,285 orthogonal angular modes is 43 seconds, which translates to only 18.8 milliseconds per angular mode. Compared to other AO-SMLM methods, CLASS-SMLM is much faster in this respect.

The exact time required for AO compensation was not clearly stated in the previous AO-SMLM papers. Marijn E. Siemons et al. claimed that REALM requires 297 frames of blinking molecules in

total ((9 different mode biases per Zernike mode) × (11 Zernike modes per round) × (3 rounds) = 297) in their paper¹. However, they did not mention the exposure time or speed of the deformable mirror. Assuming that the exposure time was similar to ours (50 ms) and ignoring the time needed for DM control, the total time is approximately (297 frames) × (50 ms/frame) = ~14.8 seconds for 11 Zernike modes. Therefore, REALM takes about 1.35 seconds per Zernike mode, which is around 71.7 times slower than CLASS-SMLM.

We added the following sentences to describe the aberration correction speed of our CLASS-SMLM.

“The recording of complex field maps takes 35 seconds, and CLASS calculation takes less than 7 seconds for an area of ~33×33 μm², the FOV in the present study. Once aberration is evaluated, it takes less than a second to generate its correction map and display it on the SLM. Therefore, the total time for 2,285 orthogonal angular modes is 43 seconds, which translates to only 18.8 milliseconds per angular mode. Compared to other AO-SMLM methods, CLASS-SMLM is much faster by one or two orders of magnitude in terms of correction speed per mode.”

As a direct application of CLASS-AO to SMLM, the manuscript contains convincing demonstrations for tissues and whole-mount animals. However, the manuscript is written in a way as an application note of CLASS. It lacks the explanations of the sensing concept and exact procedure in AO correction. How would these specific procedures affect the subsequent SMLM imaging. What type of specimen it would be mostly suited and which ones it would not. The current manuscript lacks performance characterizations (#4) and comparison with the state-of-the-art method in SMLM-AO (#5).

We would like to reiterate our gratitude for the invaluable opinions and suggestions provided by the reviewer. As explained so far, we have thoroughly addressed critical issues raised by the reviewer, including the performance characterization and comparison with other AO-SMLMs, through additional experiments and analysis. As a result, our paper has become even clearer that our research has made a significant contribution to the field of SMLM.

References

1. Siemons, M. E., Hanemaaijer, N. A. K., Kole, M. H. P. & Kapitein, L. C. Robust adaptive optics for localization microscopy deep in complex tissue. *Nat. Commun.* **12**, 1–9 (2021).
2. Mortensen, K. I., Churchman, L. S., Spudich, J. A. & Flyvbjerg, H. Optimized localization analysis for single-molecule tracking and super-resolution microscopy. *Nat. Methods* **7**, 377–381 (2010).
3. Dempsey, G. T. *et al.* Photoswitching mechanism of cyanine dyes. *J Am Chem Soc* **131**, 18192–18193 (2009).
4. Gidi, Y. *et al.* Unifying Mechanism for Thiol-Induced Photoswitching and Photostability of Cyanine Dyes. *J. Am. Chem. Soc.* **142**, 12681–12689 (2020).
5. Ovesny, M. *Computational methods in single molecule localization microscopy Super-resolution microscopy View project.* (2016).
6. Kim, M. *et al.* Label-free neuroimaging in vivo using synchronous angular scanning microscopy with single-scattering accumulation algorithm. *Nat. Commun.* **10**, 1–9 (2019).
7. Yuan, P. *et al.* Erratum: TREM2 Haplodeficiency in Mice and Humans Impairs the Microglia Barrier Function Leading to Decreased Amyloid Compaction and Severe Axonal Dystrophy. *Neuron* **92**, 252–264 (2016).
8. Sigal, Y. M., Zhou, R. & Zhuang, X. Visualizing and discovering cellular structures with super-resolution microscopy. *Science* **361**, 880–887 (2018).
9. Lelek, M. *et al.* Single-molecule localization microscopy. *Nature Reviews Methods Primers* **1**, (2021).
10. Burke, D., Patton, B., Huang, F., Bewersdorf, J. & Booth, M. J. Adaptive optics correction of specimen-induced aberrations in single-molecule switching microscopy. *Optica* **2**, 177 (2015).
11. Mlodzianoski, M. J. *et al.* Active PSF shaping and adaptive optics enable volumetric localization microscopy through brain sections. *Nat. Methods* **15**, 583–586 (2018).
12. Xu, F. *et al.* Three-dimensional nanoscopy of whole cells and tissues with in situ point spread function retrieval. *Nat. Methods* **17**, 531–540 (2020).
13. Kang, M. S., Jin, S. & Cho, H. J. MRI investigation of vascular remodeling for heterogeneous edema lesions in subacute ischemic stroke rat models: Correspondence between cerebral vessel structure and function. *Journal of Cerebral Blood Flow and Metabolism* **41**, 3273–3287 (2021).

14. Lee, E. J. *et al.* Three-dimensional visualization of cerebral blood vessels and neural changes in thick ischemic rat brain slices using tissue clearing. *Sci. Rep.* **12**, (2022).
15. Xu, J., Ma, Y., Yu, T. & Zhu, D. Quantitative assessment of optical clearing methods in various intact mouse organs. *J. Biophotonics* **12**, (2019).
16. Matryba, P. *et al.* Systematic Evaluation of Chemically Distinct Tissue Optical Clearing Techniques in Murine Lymph Nodes. *The Journal of Immunology* **204**, 1395–1407 (2020).
17. Matryba, P., Łukasiewicz, K., Pawłowska, M., Tomczuk, J. & Gołąb, J. Can developments in tissue optical clearing aid super-resolution microscopy imaging? *International Journal of Molecular Sciences* **22**, 6730 (2021).
18. Weiss, K. R., Voigt, F. F., Shepherd, D. P. & Huisken, J. Tutorial: practical considerations for tissue clearing and imaging. *Nature Protocols* vol. 16 2732–2748 Preprint at <https://doi.org/10.1038/s41596-021-00502-8> (2021).
19. Bossolani, G. D. P. *et al.* Comparative analysis reveals Ce3D as optimal clearing method for in toto imaging of the mouse intestine. *Neurogastroenterology and Motility* **31**, (2019).
20. Huff, J. The Airyscan detector from ZEISS: confocal imaging with improved signal-to-noise ratio and super-resolution. *Nat. Methods* **12**, ii (2015).
21. Kim, J. *et al.* Oblique-plane single-molecule localization microscopy for tissues and small intact animals. *Nat. Methods* **16**, 853–857 (2019).
22. Yoon, S., Lee, H., Hong, J. H., Lim, Y.-S. & Choi, W. Laser scanning reflection-matrix microscopy for aberration-free imaging through intact mouse skull. *Nat. Commun.* **11**, 5721 (2020).
23. Jo, Y. *et al.* Through-skull brain imaging in vivo at visible wavelengths via dimensionality reduction adaptive-optical microscopy. *Sci. Adv.* **8** (2022).
24. Pleiner, T., Bates, M. & Görlich, D. A toolbox of anti-mouse and anti-rabbit IgG secondary nanobodies. *Journal of Cell Biology* **217**, 1143–1154 (2018).
25. Bates, M., Huang, B., Dempsey, G. T. & Zhuang, X. Multicolor super-resolution imaging with photo-switchable fluorescent probes. *Science* **317**, 1749–1753 (2007).
26. Kwon, Y. *et al.* Computational conjugate adaptive optics microscopy for longitudinal through-skull imaging of cortical myelin. *Nat. Commun.* **14**, (2023).

REVIEWERS' COMMENTS

Reviewer #2 (Remarks to the Author):

The authors have addressed most of my concerns in their detailed response to reviews. The argument regarding novelty and differentiation with respect to prior work is well articulated in the response to reviews but that was not clearly conveyed in the manuscript. I urge the authors to rewrite the exposition in the beginning of the paper to clarify these points.

The authors point out the differences between effects of sample induced aberration vs scattering -- and they have added a sentence to clarify this. This addresses one of my concerns but I do think more could be done in highlighting this subtle difference in the introduction.

I think the comparisons with REALM could well be moved to the main paper instead of supplemental if possible.

Reviewer #3 (Remarks to the Author):

The authors have addressed each of my concerns. I am partially convinced by their clarifications of the study's proposed novelty and impact, and I note the other two reviewers are positive for the manuscript in this regard. Given these circumstances, I endorse the manuscript for publication.

Reviewer #4 (Remarks to the Author):

The authors have provided a substantial revision of the original manuscript in terms of comparison with other SMLM-AO method, further demonstrations on biological specimens and quantified substantial aberrations that exists in cleared tissues. The new data has addressed most of the technical questions from the reviewers.

Although in terms of technical advances/innovations, the revised manuscript remains lacking, the submitted work is an good example of application of an advanced adaptive AO method in super-resolution microscopy yielding significant capacity improvement in imaging depth.